# Greenhouse gas flux studies: An automated online system for gas emission measurements in aquatic environments

Nguyen Thanh Duc[1,4], Samuel Silverstein[2], Martin Wik[3], Patrick Crill[3], David Bastviken[4], Ruth K. Varner[1]

[1]Institute for the Study of Earth, Oceans and Space and Department of Earth Sciences, University of New Hampshire, Durham, 03824, USA

[2] Department of Physics, Stockholm University, 106 91, Sweden

[3] Department of Geological Sciences, Stockholm University, 106 91, Sweden

[4]Department of Thematic Studies - Environmental Change, Linköping University, 581 83, Sweden

*Correspondence to*: Nguyen Thanh Duc (thanh.duc.nguyen@liu.se)

**Abstract.** Aquatic ecosystems are major sources of greenhouse gases (GHG). Robust measurements of natural GHG emissions are vital for evaluating regional to global carbon budgets and for assessing climate feedbacks on natural emissions to improve climate models. Diffusive and ebullitive (bubble) transport are two major pathways of gas release from surface waters. To capture the high temporal variability of these fluxes in a well-defined footprint, we designed and built an inexpensive device that includes an easily mobile diffusive flux chamber and a bubble counter, all in one. In addition to automatically collecting gas samples for subsequent various analyses in the laboratory, this device also utilized a low-cost carbon dioxide ($CO_2$) sensor (SenseAir, Sweden) and methane ($CH_4$) sensor (Figaro, Japan) to measure GHG fluxes. Each of the devices were equipped with an XBee module to enable a local radio communication (DigiMesh network) for time synchronization, and data readout at a server-controller station on the lakeshore. Software of this server-controller was operated on a low-cost computer (Raspberry Pi) which has a 3G connection for remote control and monitor functions from anywhere in the world. This study shows the potential of a low-cost automatic sensor network system to study GHG fluxes on lakes in remote locations.

## 1 Introduction

Despite the fact that lakes and impoundments only cover around 3.7% of the continental area (Downing et al., 2006;Verpoorter et al., 2014), their contribution to global carbon dioxide ($CO_2$) and greenhouse gas (GHG) budgets are substantial (Tranvik et al., 2009;Bastviken et al., 2011). Lake emissions are not only large, but previous studies also highlight large uncertainties in overall emission estimates. For example, a recent synthesis of $CH_4$ emissions from northern lakes and ponds reveals that these aquatic environments contribute 16.5 Tg $CH_4$ $yr^{-1}$, equivalent to more than 65% of inverse model calculation of all natural $CH_4$ sources in the region where $CH_4$ fluxes were believed to be largely emitted from wetlands (Wik et al., 2016b). In addition, the climate sensitivity of natural emissions of GHGs is not well understood, but metadata analyses showed that $CH_4$ emissions and the ratio of $CH_4$ to $CO_2$ emissions increase markedly with the increasing temperature (Yvon-Durocher et al., 2014;Marotta et al., 2014). Previous field studies of GHG emissions are still limited in their spatial and temporal resolution, which potentially result in poorly resolved measurements and biased estimates (Wik et al., 2016a). For this reason, there is a need for new and improved approaches to study the emission of $CH_4$ and $CO_2$ from open, fresh water ecosystems at both higher temporal and spatial resolutions.

Using conventional techniques with well-defined footprints, emitted gases can be trapped in air-filled flux chambers (FC) or submerged water-filled funnels (Chanton and Whiting, 1995). When properly designed with a light weight chamber that has limited intrusion into the water surface, a mooring to enable the chamber to follow wave and water motion, the flux chamber method, which can trap both diffusive and ebullitive (bubble) fluxes, has been repeatedly shown to have negligible bias in gas flux measurement at the air-water interface relative to $SF_6$ assessments (Cole et al., 2010) or other independent non-

invasive methods (Gålfalk et al., 2013;Lorke et al., 2015). A submerged funnel moored to allow movement around a specified area can be deployed to specifically trap gas bubbles released from the sediment surface (Wik et al., 2013). For both flux chambers and submerged funnels, the trapped gas is commonly collected manually with syringes after a specific time interval, and analyzed in the laboratory. Both methods are inexpensive in terms of equipment and work well to quantify
gas emission in a defined relatively small area. However, they are labor intensive due to the need for repeated visits for both deployment and sample collection and therefore often result in low temporal resolution of emission measurements. During short term measurements, there is also a high probability of missing potentially rare and episodic ebullition events entirely. In contrast, during long term chamber or funnel deployments, $CH_4$ dissolution and or oxidation in the water that is in contact with the trapped gas could result in an underestimation of flux. Most previous measurements reported in the literature were
based on infrequent measurements within short time frames (0.5 to 24 hrs) and likely did not capture ebullition in a representative way thereby resulting in underestimation (Wik et al., 2016a). As a result, there is a high uncertainty in extrapolations and modeling of $CH_4$ and $CO_2$ emissions over time from open water ecosystems (Smith, 1985;Walter et al., 2001;Bastviken et al., 2004;Meng et al., 2012). High frequency measurements over long periods with broad spatial coverage of studied areas could reduce this uncertainty and result in more representative gas emission estimates. Regarding the
floating chamber approach, there are automated methods in which the trapped gases in the chamber can be sampled with a system of pipes and large pumps connected to a gas analyzer (Goodrich et al., 2011;Goulden and Crill, 1997). This can better address the temporal variability, but the gas analyzer equipment is typically expensive. The chambers also need to be relatively close to the gas analyzer so this method can be limited in spatial coverage.

Carbon dioxide flux measurements require a short time period for chamber deployment due to rapid equilibration. There are commercially available high precision $CO_2$ sensors available (e.g: Li-Cor, Vaisala-$CO_2$) (Johnson et al., 2010;Anderson et al., 1999) which can be connected to chambers for $CO_2$ analysis. However, their cost makes it difficult to afford many simultaneous measurements across a study area. Recently, Bastviken *et al*. (2015) proposed the use a low-cost $CO_2$ sensor and developed applications for p$CO_2$ and $CO_2$ flux measurement in outdoor environments.
High frequency measurements of the timing of ebullitive events have been made using techniques based on video/photo or hydro acoustic methods (Ostrovsky et al., 2008;Tassin and Nikitopoulos, 1995). Acoustic methods have a high potential for solving the spatial heterogeneity of gas emission, but this technique can have a high cost for equipment and there remains some uncertainty in quantifying gas emissions (Ostrovsky et al., 2008;DelSontro et al., 2015). In addition, these techniques
may work well in ecosystems with frequent ebullition, but sonar scanning can be time and power consuming over extended periods in ecosystems where ebullition is less frequent. In such systems, there is a need for inexpensive and power-efficient equipment for long term, continuous monitoring of ebullition. Varadharajan et. al (2010) developed a low-cost automated trap to measure ebullition flux using an inverted funnel connected to a pressure sensor whose signal was recorded by a commercial data logger. This type of commercial data logger and funnel still requires manual maintenance and gas release
which also means a high potential for missing ebullition events when the trap is full of gas. The eddy covariance (EC) technique is increasingly used for long-term monitoring of terrestrial and lake-dominated landscapes, but it is expensive in terms of equipment (Vesala et al., 2012;Deemer et al., 2016). In addition, EC measurements were not designed to account for any small-scale spatial variability from different types of areas that lies within the footprint of the measurement.

To increase the quality and quantity of observations of aquatic GHG emission, we developed a low-cost, simple, robust and portable device with a well-defined footprint for investigating gas flux at the water-air interface. This is a follow up from our previous open-tech published work focused on measuring $CH_4$ using an automated flux chamber (AFC) (Duc et al. 2013), now substantially improved by including sensors to reduce the need of laborious manual sampling and analyses, a wireless on-line readout-control device that has the capability to simultaneously measure ebullitive fluxes by an automatic bubble
counter (ABC) and diffusive $CH_4$ and $CO_2$ fluxes by an automated floating chamber. Taking advantage of small, low cost $CH_4$ and $CO_2$ sensors, we have modified our AFC, which is composed of a flux chamber connected to an automated control box (Duc et al., 2012), to measure $CH_4$ and $CO_2$ flux from aquatic environments. The $CH_4$ sensors tested here were Taguchi-type semi-conductor Gas Sensors sold by Figaro Engineering Inc., Osaka, Japan or Panthera Neodym Technologies, Canada (sensors described below).. Eugster and Kling (2012) showed successfully that a similar sensor (TSGS2600) has potential to
measure $CH_4$ at ambient air concentrations. The sensors have a high sensitivity to relative humidity and temperature, but

these responses can be corrected for to yield corrected $CH_4$ signals (see below). The $CO_2$ sensor used here ($CO_2$ Engine ELG K33, from SenseAir, Sweden) is a low-power module that measures $CO_2$, temperature and relative humidity. Therefore, this $CO_2$ sensor can provide temperature and humidity data to correct the $CH_4$ sensor response. The sensor-equipped AFCs were combined with submerged funnels for automated detection of bubbles (the ABC). Here, we suggest a solution to automatically collect or release the trapped gas, and restart the bubble trap by using a pump and valve system, which are controlled by an inexpensive microcontroller-based data logger, based on the feedback of the pressure signal.

## 2. Methods

In this section, we describe the technical details of our new device (a combined AFC and ABC) that simultaneously measures $CH_4$ ebullition and diffusive $CH_4$ and $CO_2$ emissions. This device operates and communicates (to receive working parameters and send data) within a digimesh network using a Xbee transmitter module (XBP24-AUI-001J, Digi International, USA). The system consists of a floating control box that houses the electronics, a floating chamber and a submerged funnel (Figure 1). The control box is a watertight case which stores a power source (either a 12V 7Ah lead-acid battery or a 12V 55Ah lithium ion battery (Power Pack LS 55, vuphongsolar.com, Vietnam), diaphragm pumps, electronic valves, a pressure sensor and the electronic controller boards inside, and had a solar panel mounted on the top of the box. The control box connects to either the chamber or the funnel or both of them. Compared with the previous version in Duc et al. (2013), the electronic controller boards, including the power control board and the data logger board, have been redesigned to include an open 5Vdc supply for a $CH_4$ sensor, an open I2C connection for a $CO_2$ sensor, and an open UART2 connection for XBEE radio communication.

### 2.1 Ebullition counter

The ABC was based on an inverted funnel design similar to (Wik et al., 2013) adopting the measurement principle of Varadharajan et Al. (2010). From the funnel stem, a 30-cm PVC pipe (10 mm I.D.) was attached to accumulate bubbles. The maximum trapped bubble volume for this system is ~ 28 mL. The other end of the PVC pipe was attached to an inverted 10 mL syringe whose tip was connected to a differential pressure sensor (26PCAFA6D, Honeywell, Sensing and Control, Canada; this sensor was chosen for being similar to Varadharajan et al. (2010) and being compatible with the electronics in our system) via a polyurethane tube (3.175 mm inner diameter, Clippard URT1-0805; Figure 1). The pressure sensor was power by regulated 10Vdc; its signal was amplified 495-fold by an AD620 chip (Analogue Devices; USA). Gas accumulating in the pipe pushes down the water level relative to the water level outside the pipe, and this water level difference generates a pressure that is proportional to the gas volume in the pipe. The ebullition rate (mL/min) is determined from the change in the differential pressure inside the pipe over time; therefore, it is important to make the trap gas tight.

The ABC can be programed to simulate the deployment cycle of a manual trap including capturing bubbles and releasing of gases when the trap is full. To enable autonomous operation for long deployment periods, not only the pressure sensor but also a pump and a 2-way valve were connected to the bubble trap via the polyurethane tube and two T-connectors (Figure S1). The pump and valve were powered by 12Vdc.

The microcontroller-based datalogger board continuously reads the amplified pressure sensor signal, and a step-wise pressure increase from gas accumulation indicates an ebullition event that is recorded with date and time stamps. The bubble measuring cycle of the ABC in the field includes initiation, measurement and ventilation stages. In the initiation state, the pump injects a small amount of air (about 5 mL) into the sampler to push any condensation water droplets out of the tube and to have a starting pressure equivalent to the sensor detection limit. During the measurement state, bubbles are trapped in the funnel and the pressure signal is continuously monitored by the datalogger. When the pressure signal increases to a threshold level indicating that the bubbles have filled up the PVC pipe, the headspace of trapped bubbles can be either vented away or measured in a connected $CH_4$ and $CO_2$ sensor box. The controller activates a ventilation cycle in which the pump purges the trap, and then the valve opens for ventilation. The valve closes again when the pressure signal drops down to the initial detection limit level. This also prevents water from entering the tube which could cause moisture blockage

interfering with sensor response. The ventilation stage cycles three times until the headspace is replaced by air. This measuring cycle (Figure S1) makes the ABC fully automated and operational over long periods - week to months or perhaps years given adequate power supply.

The pressure data can be recorded either to an SD card on the data logger or by wireless transfer to an onshore computer for subsequent transfer to a cloud server (See Section 3.3 Wireless network in Supplementary material). The data file is then processed (Matlab, etc) to extract the ebullition events from baseline noise based on the stepwise increase of the pressure signal. When the ABC was deployed in the field, the baseline noise increased. Even if the pressure sensor is pre-calibrated and has a temperature compensated range from 0 to 50°C, the weather conditions, including temperature, wind and waves,

will physically affect (shrink or expand) the bubble in the trap. Therefore, the noise removal is a critical procedure in data processing to extract the bubble events.

The regular electric noise, drift, and wind/wave effects on the pressure sensor generate high frequency low level signals. A bubble, on the other hand, will generate an abrupt jump that raises the level of pressure signal (Figure 1). In general, this

leads to periods with constant average pressure separated by a finite number of abrupt signal jumps to new pressure levels due to bubbling. This reflects a piecewise constant signal (Little and Jones, 2011). The noise in the signal needs to be removed to identify the timing and volume of ebullitive events. The classical noise removal solvers, such as smoothing, or filtering over a moving window, have several limitations when a signal can abruptly change, and these abrupt changes of pressure signals are what need to be allocated and preserved. From our field measurement data, the noise, which generally is

symmetric and tailed caused by temperature changes (Figure 1), can be removed by the jump penalization method (Little and Jones, 2011). This jump penalization solver was chosen based on the observed results from 10 different noise removal solvers that were included in a "piecewise constant toolbox" (http://www.maxlittle.net/software/). This toolbox implements algorithms for noise removal from 1D piecewise constant signals, such as total variation and robust total variation denoising, bilateral filtering, K-means, mean shift and soft versions of the same, jump penalization, and iterated medians (Little and

Jones, 2011). After the noise was removed, the denoised data is composed of flat regions at different pressure levels and the boundary of those regions. The pressure levels are proportional to the volume of bubbles in the trap and the locations of the jumps are the time when bubbles enter the trap. These events were detected by applying point-wise (1ˢᵗ order) differentiation calculations on the denoised data. The positive differentiates, with peak heights greater than three times the standard deviation of the baselines, were identified as ebullition events. A report data file including date, time of the ebullition event

and sizes of bubble were exported as a text file.

### 2.2 Measuring $CH_4$ and $CO_2$ flux in the AFC.

The AFC system presented in Duc et al. (2013), was improved by equipping the floating chamber with previously described low cost $CH_4$ and $CO_2$ sensors. To protect these sensors in high humidity environment, their electronics boards were coated by polyurethane resin (arathane 5750 or Ultifil 3000-010, details are in Section 1.1 Sensor coating in Supplementary

material). To prevent water splashing, the sensors were placed in a protected plastic box with holes for air through-flow mounted in the chamber. A detailed design is described in Bastviken *et al.* (2015), however in this study the condensation protection sheet was not used. A rubber tube (230x65 mm Inner Tube Straight Valve Stem, Part # 952932367600, esska.se) was attached to the chamber to automatically open/close the chamber for ventilation or accumulation phase by inflating and deflating the tube, respectively.


The $CH_4$ sensor was configured as shown in Eugster and Kling (2012). It is powered with 5Vdc and its analog signals are recorded via the analog input of the datalogger board (Duc et al., 2012). The $CO_2$ sensor data, including $CO_2$ concentration, relative humidity and temperature, were transferred to the datalogger via an I2C connection. The $CO_2$ sensor is powered with 10Vdc. The $CO_2$ sensors which were used in this study were prepared as described in Bastviken et al. (2015). In the recorded

data file, in addition to the time stamp and sensor data, there is a chamber open/close marker. This helps to identify the accumulation and ventilation phases of the chamber. These data are post-processed with a script (written in Matlab, MathWorks, USA) to determine the fluxes during the chamber accumulation period.

Methane flux is determined based on the change of filtered $CH_4$ sensor signals over an accumulation period. The filter is set to select data period in which the variation of RH and temperature in the chamber are small enough to not affect $CH_4$ sensor signals. The diffusive flux is estimated from the best linear increase of $CH_4$ sensor signals without ebullition event. Additional details are presented in the $CH_4$ sensor calibration section. The $CO_2$ sensor was tested previously for use in flux

chambers to determine $CO_2$ emission (Bastviken et al., 2015). The slope of the $CO_2$ concentration linearly changing in the time range, which has $R^2$ higher than 0.98, is extracted as the rate of $CO_2$ emission per time. In our field study, the chamber is closed for 100 minutes and open about 20 minutes for ventilation, and data from the sensors was output every 1 minute. The GHG flux is calculated using the following equation.

$$F = \frac{\Delta C}{\Delta t} \frac{PV}{RT} \frac{60 * 10^{-6}}{A} \qquad\qquad \text{Eq. 1}$$

where $F$ is flux (e.g. mol m$^{-2}$ h$^{-1}$), $\Delta C/\Delta t$ is change in GHG mixing ratio over time in the FC headspace (ppmv min$^{-1}$), P is atmospheric pressure (atm), V is the FC volume (6300 mL), R is the gas constant (82.0562 mL atm K$^{-1}$ mol$^{-1}$), T is the temperature (K), A is surface area of the FC (0.069 m$^2$), 60 is a conversion factor from min$^{-1}$ to h$^{-1}$ and $10^{-6}$ is a conversion factor from ppmv to fractional mixing ratio measured in the gas. The flux time unit is in hours representing a relevant time unit given the accumulation time of the chamber. As this study focuses on evaluating the sensor response to the change in

mixing ratio of $CH_4$ and $CO_2$ gases in the chamber in which the sensor signal is recorded every minute, the $\Delta C/\Delta t$ (ppmv min$^{-1}$) is used to demonstrate the response of $CH_4$ and $CO_2$ sensors.

## 2.3 $CH_4$ sensor test and calibration

We tested three commercial based TGS sensors for $CH_4$ including: TGS2611-E00, NGM 2611-E13, and a Panterra $CH_4$ sensor. The TGS2611-E00 sensor is equipped with a filter to reduce the influence of interference gases such as ethanol,

resulting in a more selective response to $CH_4$ (Figaro, 2013). The NGM 2611-E13 is a pre-calibrated module for natural gas alarms, which is also based on the TGS2611-E00 sensor. This module is prebuilt as a gas detector circuit with a standard pin connector. The Panterra $CH_4$ sensor (PN-SM-GMT-A040A-W20A-05-R0- S0-E1-X0-I2-P0-L2-J1-Z0, Panterra Neodym Technologies, Canada) which is based on a TGS2610 sensor, has been pre-calibrated by the manufacturer. We chose the specific sensor versions in dialogue with sensor company representatives, based on several criteria, including $CH_4$ specificity,

a sensitivity that was potentially high enough for our applications, price and power consumption.

The responses of the $CH_4$ sensor to concentration, temperature and relative humidity (RH) in the chamber were studied, as well as the effect of hydrogen sulfide ($H_2S$), which is a potential inference gas released from some sediments. On a water tank in the laboratory, the AFC was set to close on water surface for 100 minutes and open 20 minutes for ventilation. Water

temperature was regulated at different temperatures from 10 to 35°C. In temperature sensitivity experiment, the starting $CH_4$ concentration was atmospheric background levels (about 2 ppm), at which temperature were varied. In the calibration experiments, at different temperature levels, about 10 mL of $CH_4$ 1000 ppm was injected into the 7 L chamber every 5 minutes until the AFC activates the ventilation process. About 5 minutes after the injection, a 10-mL gas sample from the chamber was withdrawn and injected into a gas chromatograph equipped with a flame ionization detector (GC-FID) to

measure $CH_4$ concentration to be compared with sensor retrieved values. This test was repeated and, in later, the headspace gas in the chamber was circulated through a spectrometric gas analyzer (an FGGA with capacity to measure $CH_4$, $CO_2$ and $H_2O$; Los Gatos Research Inc.) for continuous $CH_4$ and $CO_2$ concentration measurements.

The $H_2S$ interference experiment was carried out by injecting different volumes (from 2 to 637 mL) of standard gas $H_2S$ 100

ppm (Duotec AS, Denmark) into the test AFC. The chamber headspace gas was circulated through a Biogas analyzer (Geotechnical Instrument, England) for measuring molecular oxygen ($O_2$) and $H_2S$. These results were analyzed using the JMP Pro software and Matlab to determine noise levels, quantitative flux determination limits, and a calibration equation.

## 2.4 Field deployment and monitoring

The field tests were performed on lakes at Stordalen Mire located in Abisko, Sweden (Wik et al., 2013). The floating control

box was tied to a buoy, which was anchored to the lake bottom. The funnel and the chamber were attached to the control box

with distance of 0.5 and 1 m, respectively. The funnel and chamber were able to freely move around the anchor point in an area of about 2 m radius.

## 3. Results

### 3.1 Bubble counter calibration experiment

Calibrating the bubble counter revealed that the pressure sensor cannot detect the first 5 mL gas in the trap due to the low accumulation pressure (Figure S2). Therefore, to reach the detection limit of the pressure sensor, the automatic bubble counter is started (prime pressurized) by pumping approximately 5 mL of air into the trap. This offset the ABC response in every measurement cycle. At pressures above this low-end threshold, the pressure sensor response showed a linear response to the volume of the gas captured in the trap. The upper threshold for a volume change that the trap can detect depends on
the length of the extension PVC pipe. The longer extension pipe, the wider linear range of the bubble counter. Therefore, the ABC was programmed to end a bubble trap period by venting trapped gas before the extension pipe is completely filled with gas.

In stable conditions in the laboratory, the square root baseline signal (baseline noise) of the bubble trap at all pressure levels
in the linear calibration curve is approximately 0.013 V. The detection limit calculated from three times the noise (0.039 V) is equivalent to about 0.8 mL. This means that our sensitivity is good enough to detect a bubble volume of 1ml – corresponding to a bubble size that has high occurrence probability in lake systems (Wik et al., 2013). In post data processing, any stepwise increase signal that is smaller than 0.04 V was therfore ignored. Field deployment data and the processed signal from a pressure sensor used to extract the bubble events are shown in Figure 1. The pressure sensor signal
measured in the trap was affected by air temperature, especially the diel temperature cycle. If there is no bubble in the trap, the pressure signals fluctuate around a certain median value (Figure 2a). Small bubbles that enter the trap, do not create a strong increasing stepwise signal that was easily distinguished relative to the background noise. However, small bubbles still raised the pressure signal median which can be detected by the jump penalization solver (Figure 2b). Even if individual small bubbles are not resolved, their combined contribution to the trapped gas will be detected as increasing average differential
pressure. The larger bubbles (around 3 - 4 mL) made clear stepwise increases in the pressure signal beyond the background noise and the jump penalization solver was able to extract the median of this stepwise pressure level. Larger bubbles therefore were determined with better resolution. Two cycles of bubble accumulation over a long-term field deployment from two tested devices shows that a wide variability of bubble sizes can enter the trap, from small bubble sizes (1 - 2 mL) who's signal is buried in the noise, to a large bubble (> 6 mL) that creates a large pressure signal compared to the
background (Figure 2). If bubbles entered the trap and were large enough to activate the venting mechanism during a non-logging period (in the present system this would require a total bubble volume > 28 ml), it was missed in the logged data file. However, no single bubble events larger than this threshold were experienced in the field tests so far.

### 3.2 AFC CH₄ sensor calibration experiment

The response of the $CH_4$ sensor, when the chamber closed on water surface, to changing temperature, RH, and $CH_4$
concentration (around 2ppmv), is shown in Figure 4. In the first few minutes after chamber closure, the temperature and RH changed quickly in the chamber, causing a drift in the $CH_4$ sensor signal, but once temperature and RH stabilizes, the sensor responded in a predictable way to changes in $CH_4$ concentration inside the chamber. After temperature and RH stabilization occurred in the chamber, we determined the detection limit of our instrument for $CH_4$ fluxes based on the noise of the $CH_4$ sensor. In these blank experiments, the RH was always in the range of 60 - 90%, which is within the sensor RH operating
range. The operation of the $CH_4$ sensor includes heating it to a high temperature to detect combustible gases, therefore, the temperature inside the sensor box is always higher than the water temperature. The temperature sensor of the ELG $CO_2$ sensor measured the changing of temperature inside the sensor box over the water temperature. The noise over a whole accumulation period (100 minutes) was about 2.44 ± 1.21 mV. A minimum accumulation rate limit is calculated as five times the noise or about 12.2 mV. Therefore, we calculated the $CH_4$ concentration increase that generated a $CH_4$ sensor signal

of 12.2 mV to be equivalent to 5.25 ppm and used this to calculate a minimum detectable $CH_4$ flux. In an accumulation period, the accumulation rate detection limit of this sensor embedded in the chamber is 5.25 ppm per 100 minutes (0.0525 ppm per minute).

At all temperatures (10, 20, 25 and 30°C), the three $CH_4$ sensor signals were well correlated to $CH_4$ concentration but these correlation lines had different intercepts depending on water temperature (Figure S3 a-c). The absolute concentrations measured by the sensors were affected by temperature and RH and were not suitable for use. To study $CH_4$ flux, we instead used the relative change of $CH_4$ concentration over time from five minutes after chamber closure to avoid the periods with the largest changes in temperature and RH. Accordingly, the differential $CH_4$ sensor signal (d0_$CH_4$sens), which is the

difference between the current measurement and the initial measurement point 5 minutes after the chamber closed on water surface, was used instead of the raw output signal from the sensor. Indeed, the differential $CH_4$ sensor signal was less sensitive to temperature and had a linear response ($r^2 = 0.98$; $p < 0.001$) across the studied temperatures (Figure 5). In this test, the variability in the temperature and RH were in ranges of 2°C and 5%, respectively, while $CH_4$ concentration increased from atmospheric or about 2 ppm to 25 ppm. The standard least square fit model was applied on d0_$CH_4$sens as a model

response and the changing $CH_4$ concentration, temperature and RH as model effects. The result showed that the variability of temperature ($p = 0.038$) and RH ($p = 0.867$), have less influence on the $CH_4$ sensor response compare to the $CH_4$ concentration ($p\_value < 0.001$). This reveals that these $CH_4$ sensors can be used to measure $CH_4$ flux when the temperature and RH conditions are stable in the chamber.

The effect of temperature and RH can be corrected for in the response of the sensor using an algorithm developed by Eugster and Kling (2012), but this was not applied in our study because we were not able to simulate the natural variations of outdoor temperature and RH conditions on our control experiment. Therefore, periods with stable temperature and RH were used and the calibration curve for the TGS2611-E00 $CH_4$ sensor in our application was the average linear response of d0_$CH_4$sens versus the changing of $CH_4$ concentration (d0_$CH_4$conc) without temperature and RH correction (Figure 5)


$$d0\_CH_4 sens \ = 1.256 \times d0\_CH_4 conc \ + \ 5.871 \qquad Eq. \ 2$$

where d0_$CH_4$sens is the voltage change of the $CH_4$ sensor in mV, and d0_$CH_4$conc is in ppmv. Comparing with this sensor, calibration results showed that the pre-calibrated Figaro NGM2611-E13 module has about the same response to the change

in $CH_4$ concentration at all temperatures. The NGM2611-E13 had a regression equation as following:

$$d0\_CH_4 sens \ = 1.116 \times d0\_CH_4 conc \ + \ 1.771 \qquad Eq. \ 3$$

The Panterra $CH_4$ sensor showed a different response at different temperatures (Figure 4). Its calibration lines had different

response at 10 and 15°C, and its d0_$CH_4$sens has negative response when d0_$CH_4$conc is higher than 15 ppm at 20 and 30°C.

In the $H_2S$ interference test, the injected volume of $H_2S$ standard increased from 2 to 637 mL. The Biogas analyzer did not detect any $H_2S$ even when the estimated $H_2S$ concentration in the chamber was 9 ppm. This level is close to the detection limit of the instrument, and given the minimum analytical uncertainty of $\pm$ 10 ppm, it is likely that $H_2S$ was present in high

enough amounts to affect the $CH_4$ sensors. During the $H_2S$ addition, the $CH_4$ sensor signal increased to more than 5 times of baseline noise; therefore, $H_2S$ was considered to affect the sensor response, in agreement with sensor producer tests.

**3.3 $CH_4$ and $CO_2$ flux with the AFC**

The pilot field deployment of the AFC embedded $CH_4$ and $CO_2$ sensors showed that the system was effective for measuring the variation of $CH_4$ and $CO_2$ concentration in the chamber over time (Figure 6 a-c). The automatic mechanism developed to close the chamber for flux measurements and open the chamber for ventilation periods/phases helped to reduce condensation and allowed for a linear response of the $CO_2$ sensor (Figure 6b). This is an improvement over past work and allows for the sensor to be deployed in the field for long time periods. There was a situation when the chamber was closed on water surface

for a whole night due to a low-battery. As a result, the saturated RH in the chamber became higher than 100% and caused condensation and malfunction in the sensor until drying (discussed in Bastviken et al. 2015). After two ventilation cycles, the $CO_2$ sensor dried and the baseline decreased to the normal linear response range. The $CO_2$ sensor responses were not affected by temperature and RH in our experiment. Therefore, $CO_2$ flux is determined from the slope of the best linear response data in an accumulation period.

During measurement periods, right after ventilation, $CO_2$ concentrations in the chamber are supposed to be equal to the atmospheric $CO_2$ concentration above the lake surface. These initial $CO_2$ concentrations varied within a range of 516 - 1179 ppmv with higher mixing ratios during nighttime. Because the chamber ventilation time was early in the development adjusted to allow complete ventilation of the chamber headspace, the elevated starting concentrations may reflect actual concentrations if stable atmospheric conditions resulted in a near-ground buildup of $CO_2$ released from the lake and the surrounding mire ecosystem.

The field deployments revealed that there were many periods in which temperature and RH conditions of the chamber were stable enough (Figure 6a) for the Figaro $CH_4$ sensors to adequately measure the changing of $CH_4$ mixing ratio in the chamber. In cases where temperature and RH varied a lot, the data processing script determined periods of data where the variation of temperature and RH was less than 2°C and 5%, respectively, defining periods for which $CH_4$ sensor data could be reliably evaluated. If ebullitive $CH_4$ entered the chamber headspace, there was a clear positive change in the sensor signal output. This was easily identified as a stepwise increase of the $CH_4$ sensor signal over a very short time. This signal identified the type of ebullitive flux that could be measured over that chamber closure period. For diffusive $CH_4$ flux measurements, the d0_CH₄sens data, with a sensor response of less than 30 mV (within linear calibration range) and without a stepwise jump, were scanned for a data range with best linear adjusted R square. For diffusive flux estimation, the $\Delta C/\Delta t$ (ppmv min⁻¹) in Eq. (1) is calculated using the last d0_CH₄sens point in this linear range, in which d0_CH₄sens is converted to d0_CH₄conc (ppmv) follow Eq. (2) and its coordinate in time since the chamber closed is calculated the measurement period (minute). The manually collected gas samples in the field and the $CH_4$ mixing ratio change over time (ppmv/time) in the chamber headspace determined from the sensor response showed a strong linear relationship with a deviation of less than 15% (Figure 7). The ebullitive $CH_4$, which detected by the $CH_4$ sensor in the AFC, was not concentration quantified in this study focusing on the relative changes of the methane sensor in the low range, because $CH_4$ sensor response to ebullition events was usually out of the linear calibration range.

## 4. Discussion

### 4.1 Automated ebullition measurements using pressure sensors

Deploying pressure sensors to determine the timing of ebullition events and to measure the bubble volumes has been thoroughly tested by Varadharajan et. al. 2010. Our bubble trap introduces a way to automatically reset the system after being full of gas, that allows for long-term deployment with minimum maintenance effort. This design idea is similar to the automatic bubble traps in Maeck et. al. (2014). Further, via the jump penalization noise removal method, bubble events can be detected despite the noise caused by changes in air temperature affecting the bubble volumes and therefore the differential pressure . It is important that the ABC is gas tight. This is not a simple requirement; especially because the trap is built from plastic materials meant to be easily disconnected for portability and is exposed to the outdoor environment. After a long deployment time, leaks were occasionally observed at the assemble joint of the pressure sensor. So far, if a trap leak occurs, the pressure is lower than the priming pressure threshold, which was set to trigger a warning indication to the host server controller on the lakeshore. It can be fixed by applying glue on the leak site. The detection limit of the differential pressure measurement, in our case corresponding to 1-2 ml gas, depends on the shape of the cylinder where the bubbles accumulate (Maeck et al., 2014;Varadharajan et al., 2010). Therefore, the longer and narrower a cylinder, the lower the detection limit. This leads to a trade-off, where the more sensitive systems become too tall for deployment in shallow waters, which often have proportionally higher ebullition rates (Wik et al, 2013). Our detection limit was chosen to allow deployment in shallow water with a trap height of about 0.5 m. The recent study using optical sensors in an open path funnel (Delwiche and

Hemond, 2017), that makes a shorter trap, suggests an alternative and interesting design for ebullition studies, which could be combined with the present sensor approach to also quantify $CH_4$ content in the bubbles.

## 4.2 Automatic measurement of $CH_4$ and $CO_2$ during chamber fluxes

In our application, the low cost $CH_4$ and $CO_2$ sensors can be used to measure changing $CH_4$ and $CO_2$ concentrations. It is a
direct approach to measure $CH_4$ and $CO_2$ flux from a defined-footprint area on the time scale of minutes-hours, extending over long-time periods given a suitable power supply. The chamber captures both ebullition and diffusion fluxes. Ebullition events are marked by abrupt changes in the response of the $CH_4$ sensor and therefore can be identified readily. The diffusive flux is identified by the gradual change in $CH_4$ and $CO_2$ concentration over time. We did observe ebullition events in the chamber during deployment periods, in support of the previous indications that ebullition typically accounts for a large share
of the open water flux (Figure 6c). However, since we did not calibrate the sensor for high concentrations, we could not determine the flux rate observed during these events. This remains a challenge for future work.

To study diffusive fluxes, it is important to measure the change of gas concentrations during a short period of time right after the chamber closes. This requires a gas sensor that can measure at near ambient gas concentrations. The $CH_4$ injection
experiment showed that both of the Figaro $CH_4$ sensors have sensitivity at low ppm mixing ratios and yield a linear response from ambient at about 2 ppm up to 25 ppm. The TGS2611-E00 and NGM2611-E13 have small differences in their response (slope) in the linear range, however, their responses to experimental conditions are overall very similar because they use the same sensor base. In outdoor field conditions, after closing on the water surface, it takes some time for temperature and RH in the chamber to stabilize. The rejection of data from the initial 5-minutes of measurements is important to select an initial
$CH_4$ sensor data point when the temperature and RH has become more stable to minimize the influence of these confounding factors on the relative change of $CH_4$ sensor signal. An obvious data interpretation improvement would need to modify the length of the initial period during which data is not used to the actual time it takes to reach stable enough relative humidity and temperature, instead of having the static 5-minute period used here for simplicity.

It is possible that the sensors can be used outside the range reported here by developing other calibration curves. In any case, we recommend to adjusting the AFC accumulation time to the effective range of the sensor. Alternatively, flux calculation can be based on data within the linear range only in the post-processing of the data. Calibration for the Figaro $CH_4$ sensor is recommended for each individual sensor. The response slopes of different sensors could deviate up to 12%. For practical reasons, if flux estimation with tolerance ±20% error is accepted (Wik et al., 2016a), one general calibration line can be
obtained from a calibration of at least five $CH_4$ sensors for statistical representativeness. In our study, the calibration line was obtained from the calibration experiment of eight sensors. Due to the effect of temperature and RH, the calibration curve should be based on calibration data at different water temperatures that span anticipated field conditions. Compared to Duc et al., 2012 which used the Panterra $CH_4$ sensor, the Figaro $CH_4$ sensor gave more reliable and robust flux measurement result under field conditions.

The $H_2S$ interference test revealed that $H_2S$, a corrosive gas which can be released from anoxic sediments in sulfur rich systems, may interfere with sensor response. Therefore, extra care and thorough data validation is suggested when applying the sensors in sulfur rich environments. In addition, this $CH_4$ sensor response is based on reaction between $O_2$ in air and reductant (flammable) gases; therefore, any change in concentrations of either $O_2$ or reductant gases could interfere the signal
of the sensor. This $CH_4$ sensor can combust a small amount $CH_4$ gas (about 0.0041 ppmv per minute), which needs to be considered when the $CH_4$ flux is low (near the detection limit 0.0525 ppmv per minute of the sensor in our application) and chamber accumulation time is very long.

One limitation of the $CH_4$ sensor is its power consumption. While the $CO_2$ sensor can be activated once per minute (or at
other desired time intervals), the $CH_4$ sensor needs to be heated at all times. In our case, these systems were deployed at high latitudes in the summer and the battery was recharged by a 13W solar panel. If the weather was cloudy for four to five days in a row, the battery voltage fell below 10.5 V. At this point, the system automatically turns off until the battery is recharged.

In 2017, the replacement lithium ion battery (12V 55Ah; Power Pack LS 55) helped to keep the system working continuously during longer time periods and reduced the weight of control box.

Over our deployment time, there were several chambers that were either submerged or turned over. The chambers were submerged because the rubber inner tube degraded due to UV exposure over a long period of time, generally about two sampling seasons. This problem was solved by covering the inner tube with aluminum foil or by changing to the gas delivery flow scheme shown in Figure S9 ("no sink AFC"). With this new flow design, the air in the floating control box was pumped into the rubber inner tube until the inner tube is full. When the pressure inside the rubber inner tube is more than 1 psi, the check valve opens, the excess air is blown into the chamber to refresh its headspace. Compare with previous gas flow design, in which the air is withdrawn from the chamber, this new gas flow scheme will prevent under pressure built up in the chamber during the ventilation process; hence even in a situation where the chamber cannot open due to failure of the rubber inner tube the chamber will not sink. With this configuration, the sample array presented in Duc et al (2013) cannot be used. The strong correlation between grab samples and the sensors (Figure 7) allow us however to capture the high temporal fluxes and skip the labor-intensive process of analyzing grab samples. Manual grab samples should however be taken periodically as a cross check of the sensor response. The other problem of chamber flipping was caused by wind suddenly change its direction during chamber ventilation process. To prevent this flipping, the opening side of the chamber was attached with two floating anchors (called anti-flipping anchor) (Figure S10). With this improvement, there have been no chamber flip during tests with maximum wind speeds of about 7 m/s.

## 4.3 Challenges when networking measurement systems remotely

One goal of our project was to develop an active wireless sensor network in which a small low-cost Raspberry Pi computer on the lakeshore communicates with many flux chambers and bubble traps (called clients) by Xbee radio transmitter modules 2.4 GHz. The communication is to synchronize real date time, working parameters, to check client status and to receive data from clients. The sampling sensor data rate, so far, is constrained to maintain the digimesh network working with minimum labor effort. Under harsh weather conditions (rain and hard wind), radio communication is easily broken, and then it can take some minutes to re-establish depending on the distance and number of the clients. During this offline period, the limited memory buffer of the client data logger did not allow a very high measurement frequency. As a result of this limitation, one-minute data sampling interval was used to provide stability for long-term deployment. In spite of our low measurement frequency, the results show that the system is still able to capture relative changes in $CH_4$ concentration adequately. This situation is different from applications that aim for accurate absolute concentrations in ambient air, and for such applications a high measurement frequency is more important to cancel out sensor noise in data processing than in our flux chamber application. In our AFC system, higher frequency data, can be recorded in a local 2 GB SD memory on the datalogger of each trap if desired.

In our study, the traps were on the lake surface which was usually lower than ground level and surrounded by tree and plants. Over the study season, the growth of vegetation on the lakeshore can potentially block the line-of-sight between the host controller and the traps on the lake, which can hamper radio communication. To guarantee radio communication, at least one client (i.e. chamber system) was placed on a strategic location which had a clear line of sight to the host controller. Within the digimesh network, XBee modules can form a self-configuring, self-healing wireless peer-to-peer network with other data loggers in radio range. Therefore, the host controller doesn't necessarily need to have direct line-of-sight communication with all of the traps on the lake surface. If some of the traps are out of the controller's direct range, they should be automatically passing their messages through closer clients. Therefore, it is important to keep a robust network topology.

Occasional errors within this network still occur, probably due to high humidity environment around the clients and variable weather conditions, temperature and humidity (Luomala and Hakala, 2015). This could cause failure in transfer of some initial data packets in the data file or break the communication with clients. Hence, software for this system was developed to address these errors. For example, all the data packets were encoded, so the missing data can be easily identified in the

post data processing and the host controller keeps searching to reestablish communication with "lost" clients. Details of wireless communication protocol and host-controller design are presented in the Supplementary material.

## 5. Conclusions

Resolving diffusive and ebullitive GHG fluxes at the air-water interface in a well-defined footprint area is needed so that we can accurately represent open water bodies likes lakes and streams in global $CH_4$ and $CO_2$ budgets. With the benefit of low-cost technology, we have modified simple flux chambers and bubble traps to function automatically with wireless remote monitoring and control via an internet browser. These traps are equipped with not only the sensors to monitor the fluxes in high temporal resolution but also the electro-mechanical hardware to do complex actions in the field such as venting traps and collecting gas samples (if needed). This is our first attempt to integrate several low-cost technologies to make a device to measure GHG emissions from lakes with the data updated online in real time. This device, as an open source technology for non-profit academy study, can hopefully contribute to studies of GHG emission from aquatic environments in remote and logistically difficult areas.

## Code availability

## Data Availability

## Supplement link

(from Copernicus)

## Author contributions.

NTD: hardware, software designed the AFC-ABC system, conducted the study, developed Matlab script and processed data and co-wrote the manuscript; SS: hardware, core-software designed the wireless datalogger, webserver and contributed to the manuscript; MW conducted the study and contributed to the manuscript; PC, DB, RKV gave advice through all stages of the study, contributed funding, and co-wrote the manuscript. All authors discussed the results and commented on the manuscript.

## Competing interests.

The authors declare that they have no conflict of interest.

**Acknowledgements**

This research was supported through a postdoctoral fellowship funded by the University of New Hampshire. The field deployments 2014 and 2015 were supported through two US National Science (NSF) grants to Ruth Varner: MacroSystems Biology (EF# 1241937) and the Northern Ecosystems Research for Undergraduates program (NSF REU site EAR#1063037); and by funding from the Swedish Research Council VR to David Bastviken and Patrick Crill (Grant no. 2012-00048). Work time for data analysis and manuscript preparation was also financed by VR (grant no. 2016-04829), FORMAS (grant no. 2018-01794), VINNOVA (grant no. 2015-03529), and by the European Research Council (ERC) under the European Union's Horizon 2020 research and innovation programme (grant agreement No 725546).

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

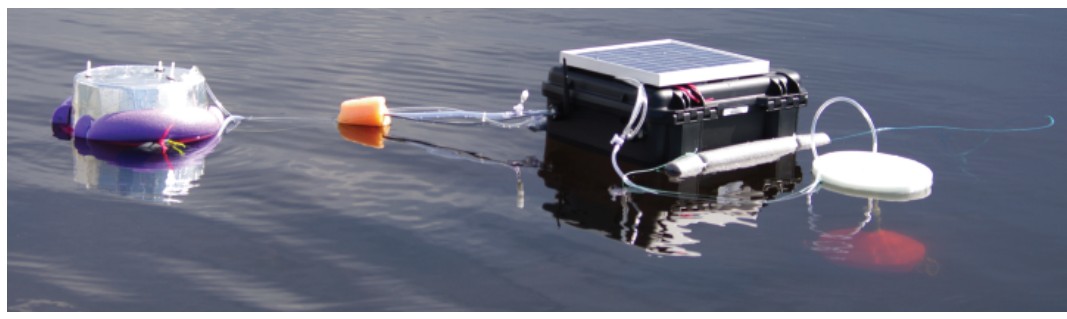

**Figure 1: The photo of a deployed AFC_ABC device consists of a floating control box that houses the electronics, a floating chamber and a submerged funnel.**

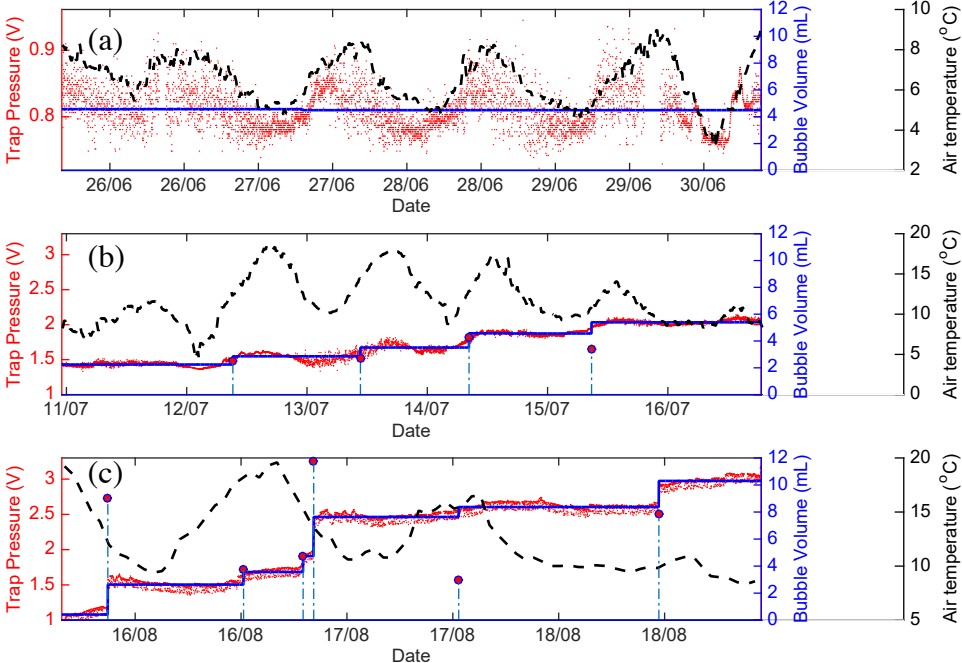

**Figure 2: ABC pressure signal and air temperature over three field deployment periods. Red dots are trap pressure signals, blue lines are the denoised pressure sensor signal, stem plots (vertical dashed lines with red circle on top) are bubble events which were detected from the stepwise increase of denoised signals, and black dash are air temperature. (a) Sample period with no bubbles entering the trap, (b) Sample period with small bubbles entering the trap, and (c) sample period with both big and small bubbles entering the trap.**

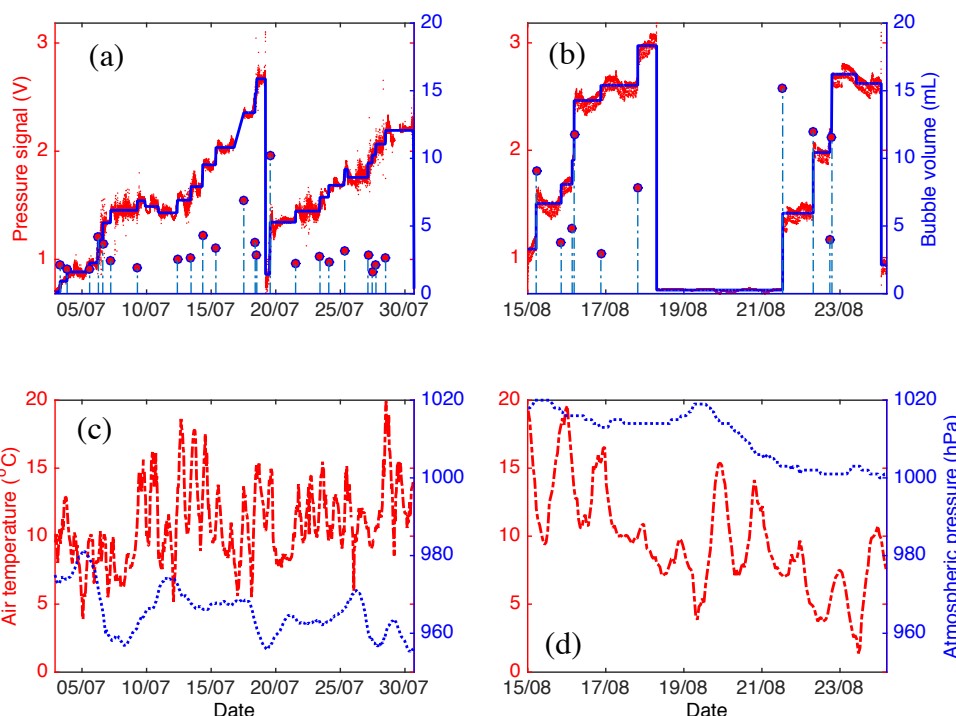

**Figure 3: Two measurement periods of an ABC deployed in Mellersta Harrsjön, Stordalen Mire, Abisko in 2015. (a and b) Sample period when bubbles entering the trap were detected from the denoised pressure signal, red dots are trap pressure signals, blue lines are the denoised pressure sensor signal, stem plots (vertical dashed lines with red circle on top) are bubble events. (c and d) Air temperature and atmospheric pressure from an onshore weather station during the same sample period, and red dash are air temperature and blue dots are atmospheric pressure.**

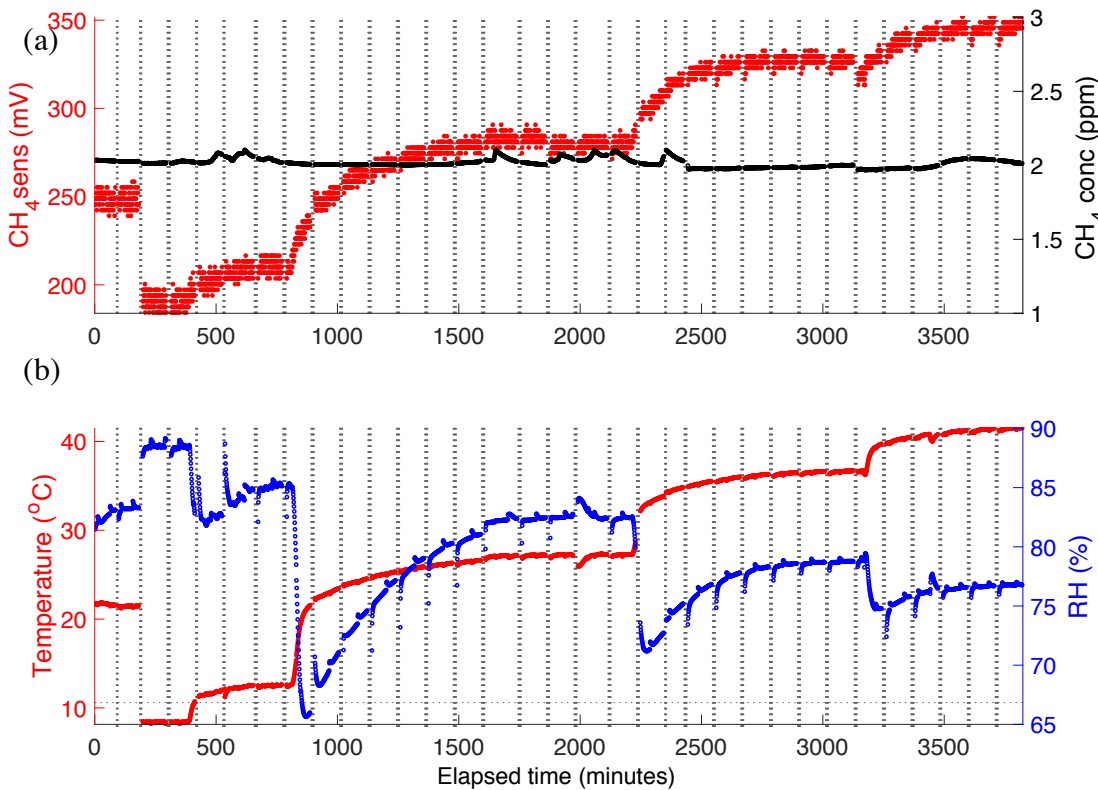

**Figure 4: Methane (mV response and concentration), temperature (°C), and RH (%) sensor responses in the initial phase of the sensor test experiment in which temperature of water tank was regulated in range from 5 to 35°C. a) CH₄ sensor signal and actual CH₄ concentration around 2ppm, and b) Temperature and RH in the chamber over the experimental period. Vertical dotted lines denote periods when chamber was opened for ventilation.**

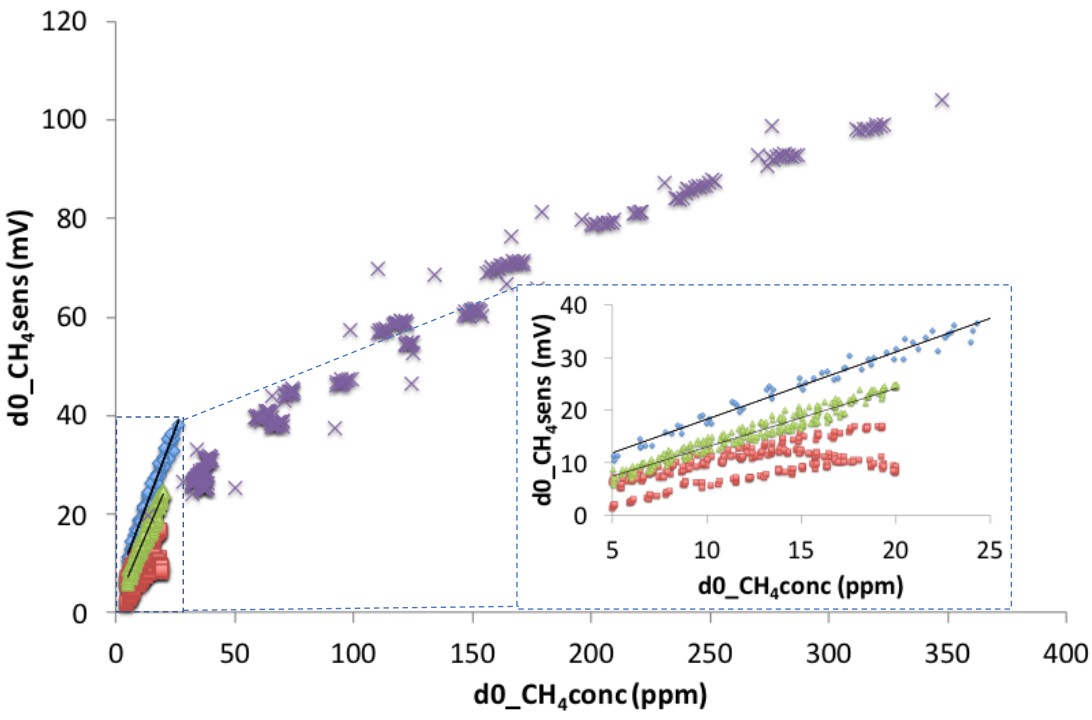

**Figure 5: Response curves of the CH₄ sensors responses at all experimental water temperatures from 10 to 30°C versus the changing of CH₄ mixing ratio measured by GC-FID or an Los Gatos Research FGGA greenhouse gas analyzer (d0_CH₄conc). The blue diamonds, green triangles and purple crosses (Δ, x: for CH₄ concentration higher than 25 ppm), and red squares represent the change in the CH₄ sensor signal over time from the chamber closure (d0_CH₄sens) of the TGS2611-E00, NGM2611-E13 and the Panterra sensor, respectively. See text for details.**

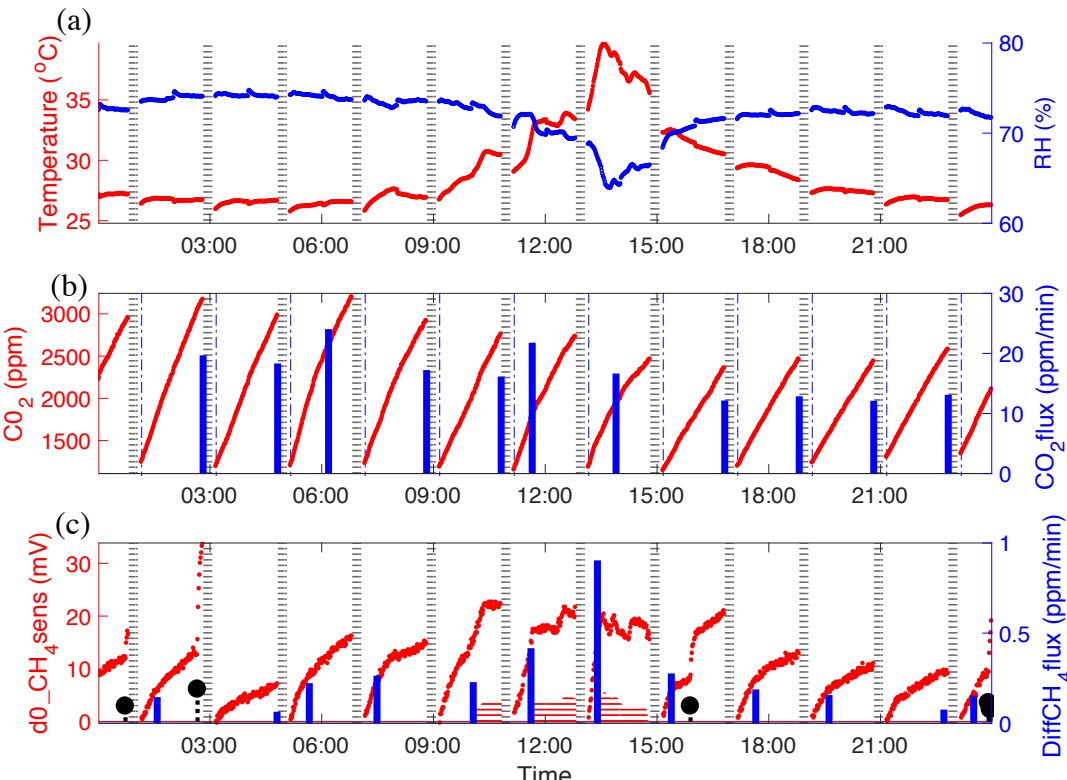

**Figure 6: Example of one day of automatic flux chamber (AFC) measurements covering 11 accumulation periods. (a) Scatter plot of temperature (left axis, red) and RH (right axis, blue) in the chamber. (b) Scatter plot of CO₂ concentration measured by an ELG CO₂ sensor (left axis) and bar plot of CO₂ fluxes calculated from slopes of the changing CO₂ concentration in time range marked from the vertical dash dot line to bar plot location (right axis). (c) Scatter plot of CH₄ sensor signal (left axis) and bar plot of CH₄ fluxes (right axis) calculated from the best linear data range when d0_CH₄sens values are in the calibration linear range (less than 30 mV), temperature and RH changes are less than 2°C and 5%. Red shaded periods indicate sampling when temperature and RH are affecting the gas sensor response and therefore these data are not used in the flux calculation. In the event of an ebullition event, the flux calculation is made with data taken prior to that event. Ebullition events are marked by the black stem plot.**

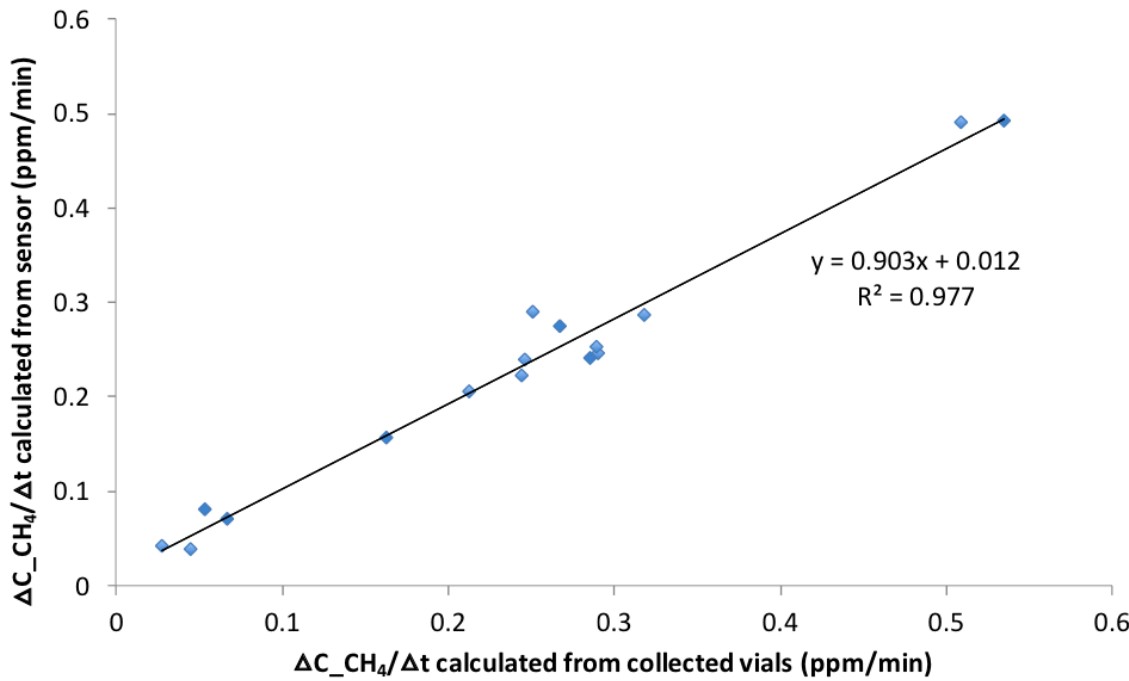

**Figure 7: Methane accumulation rates calculated from an NGM2611-E13 CH₄ sensor signal compared to accumulation rates calculated from CH₄ mixing ratio in gas samples collected at the start and end of accumulation periods.**

