# Peer review of "Greenhouse gas flux studies: An automated online system for gas emission measurements in aquatic environments"

_Hydrology and Earth System Sciences, 2019_

## Referee Comment (RC1) · Werner Eugster (Referee) · 6 May 2019

**1   Introductory remark**

According to the Steps of the HESS process of peer review, I expected the following from stage 2, Access review:

*The editor is asked to evaluate whether the manuscript is within the scope of the journal and whether it meets a basic scientific quality and if it contributes something new to the field of hydrology. They can suggest technical corrections (typing errors, clarification of figures, etc.) before posting in HESSD.*

Unfortunately this step was skipped and thus I am a bit annoyed by the sloppy presentations with plenty of technical issues that should have been solved before exposing the manuscript for discussion. Since I do not feel this to be my task I will simply dump technical remarks at the end and ask the Editor to make sure that step 2 is following after step 4 (open discussion) to address the "technical corrections (typing errors, clarification of figures, etc.)". In this case it should have been done.

**2   Scientific contents**

The authors present their newest developments of an automated online system for gas emission measurements over water surfaces, which uses low-cost sensors for $CO_2$, $CH_4$ and the necessary meteorological variables. From the pressure sensor they deduce ebullition flux, and with a smart solution using an inflatable tube to open the chamber for venting they provide a solution for automatic long-term deployments.

Thus, this is quite innovative work and my review tries to honor this, although some details were not well described (or are confusing) for a reader like me who would like to understand the details. In any case, after having gone through step 2 (all technical issues rectified), and moderate revisions, I think this paper should be acceptable for HESS.

I explicitly appreciate the technical details with 3-D printing code etc. which is something the technicians I work with definitely find useful information that they could potentially profit from.

**2.1   Major Issues**

1. The authors use three low-cost sensors, all from Figaro Inc.: the TGS2611-E00, the NGM2611-E13 (which uses the same TGS2611-E00), and the Panterra from

Neodym (I assume), which in the version I used had a percursor version of the TGS2611 or so built in. I think the authors should more clearly specify (a) what sensor the Panterra uses (and provide the company names of all sensors), (b) clarify that these are (most likely) all the same sensors in different configurations (as it reads now the reader could be getting the impression that three different sensor types were tested, which is not the case)

2. I am concerned about the low voltages that the TGS2611-E00 gives, ca. 18–35 mV according to Fig. 3. The TGS2600 that I use delivers 400-600 mV for ambient conditions, and when I look at the specifications it appears that the manufacturer considers the TGS2611-E00 to be useful in the range of 300–10,000 ppm $CH_4$ which is way above ambient range (the TGS2600 is shown with a sensitivity to $CH_4$ in the range 1–100 ppm $CH_4$ that's the reason we selected this one for near-ambient measurements in Eugster & Kling, 2012). Now there are some other publications that show that the TGS2611-E00 is actually sensitive also to near-ambient conditions, but I am not yet convinced that this is the best choice for your application given the low $CH_4$ concentrations well below the range indicated on the technical specification sheet of the manufacturer. Some more critical discussion on the sensor selection would be required in my view.

3. The authors only sample data every minute, which I find utterly coarse. They may have a reason for this, but in my own tests with the TGS2600 a one-minute measurement interval **in combination** with a 5-minute data rejection after chamber deployment (page 6, line 26; this information should actually have been given in the Methods section already, because this is an essential flaw in the system in my view) I would have lost all the information relevant to chamber fluxes (see graph below and description of unpublished internal example graph from my experiments at Toolik Lake, Alaska, USA). Thus, the authors should more precisely describe their method and critically discuss such shortcomings to help others to do better.

4. The ebullition (bubble) counter is quite interesting, but with a bubble volume of 3–4 mL required to actually leave a defensible signal, this does not yet seem to be an optimum choice. Here, a reference to and comparison with the (commercial) system of Andreas Mäck (doi:10.5194/bg-11-2925-2014) would be helpful. Since the Varadharajan et al. (2010) reference (pages 2,3,8, lines 30,22,21) is not listed in the References, I could not convince myself that this bubble counter system is really thoroughly tested and reliable. In the discussion you only say "For a long term solution, the recent study using optical sensors in an open path funnel (Delwiche and Hemond, 2017) 30 suggests an alternative and interesting design for ebullition studies, which could be combined with the present sensor approach to also quantify CH4 content in the bubbles." – thus does this mean that you are satisfied with the performance for short-term investigations? I am not really convinced and would appreciate a somewhat clearer statement what your recommendation is for studies that are shorter than a "long-term solution".

5. Your regressions (I assume you use ordinary least-squares regressions) are not correct from a statistical viewpoint (Figs 4 and 5): you must reverse the dependent and the independent variable: you want to find out how to use the signal to compute the true concentration using your regressions, not the other way round (i.e. to predict the signal based on knowledge of the concentration – that's what your regressions show).

6. I always use a fan in chambers, you don't. I understand that this corresponds to some static chambers that people use with syringe sampling, but in your case I am concerned that without a fan to mix the volume of the chamber the $CO_2$ (which is heavier than air) starts to accumulate above the water surface, and then a steep gradient creeps upwards where I expect your sensors; hence this linear increase in Fig. 5. Contrastingly, $CH_4$ which is lighter than air, quickly would accumulate under the top of the chamber, and hence probably the curvature although I would have expected that the $CO_2$ saturation should occur earlier than

the CH$_4$ saturation in such a chamber. Please comment on this and justify why not to mix the air inside the chamber to ascertain representative concentration measurements inside the flux chamber.

7. According to the manufacturers information the sensor resistance **decreases** as the CH$_4$ concentration **increases**, thus in principle the voltage you measure should **decrease** not increase with increasing CH$_4$ – but your measurements in Fig. 4 show the exact opposite of what one would expect from the manufacturer information. Do you have an explanation for this? I must admit that on short timescales I see the same (see Figure inserted below), but on longer timescales I see what I would expect from the manufacturer's data sheet. Our procedure suggested by Eugster & Kling (2012) solve this issue with the calibration – after linearisation the use of a high and low calibration point simply reverses the sign if the response is of the kind that you show in Fig. 4. If you have an explanation why the TGS2611-E00 has increasing voltage with increasing CH$_4$ concentration then this would be a helpful insight for the reader. If you don't have an explanation, maybe you have an opinion?

**2.2 Minor Issues**

1. p2/33: "The eddy covariance (EC) technique is increasingly used for long-term monitoring, but it is expensive in terms of equipment and is still being evaluated for aquatic environments." – what do you mean with this statement? I don't consider this to be correct, the method is in use beyond evaluation. Please provide some references and reword the second part. For example, we have authored a couple papers and also written a chapter in the book Eddy Covariance: A Practical Guide to Measurement and Data Analysis (chapter on lakes: doi:10.1007/978-94-007-2351-1_15). Thus the method is established (at least better than your chambers, to be more direct) – but I agree that it is costly and I agree that such low-cost sensors are important

2. p2/45: " The CH4 sensor tested here is a Taguchi Gas Sensor (TGS) (Figaro Engineering Inc., Osaka, Japan). It is a high sensitivity CH4 gas sensor...":
I completely disagree, it is a **low-sensitivity sensor** which (in the version you use) only has a manufacturer specified lower measurement range of 300 ppm $CH_4$! I already realize that our more cautious wordings about the TGS2600 (which has a higher sensitivity than the TGS2611-E00) is ignored by some others, which can lead to frustration. Be clear that this is experimental work trying to squeeze the tiny bit of information out of a sensor that is **not** made for ambient concentrations – but I agree that it has some value for such measurements.

3. You never specified which pressure sensor you used, thus it is unclear to me why you did not use an I2C sensor, there plenty of those on the market. What is the special advantage of your pressure sensor that requires an AD620 amplifier to be useful? This remains obscure to the reader.

4. Eq. 1 should use SI units or at least the same units of the same physical quantity and not include obscure conversion factors. Thus, you must decide whether your time variables should be in hours or in seconds (the primary SI unit) or minutes, please no mixtures.

5. p5/25: check your instrument information, most likely this was an LGR FGGA (not a DLT-100, which as I remember is a $CH_4$-only instrument) that measures $CH_4$ and $CO_2$.

6. p6/8: you did not specify what your "baseline noise" actually is. Is it the square-root of the variance or the noise baseline derived from an Allan variance plot, or anything else? Some more details in the Methods section would be really helpful.

7. p6/13: "The pressure in the trap was affected by air temperature, especially the diel temperature cycle." – this sounds like an error, pressure is a physical entity that is independent of temperature, thus this must be wrong. What I can imagine

is that you mean that your pressure sensor signal (but not the pressure itself) depended on air temperature. Please correct.

8. p8/39: "However, since the sensor is not calibrated for very high concentrations, we could not determine the flux rate observed during these events." – I completely disagree, at least for $CH_4$ (you do not really reveal any necessary details on the $CO_2$ measurements . . . ): the TGS2611-E00 has a specified measurement range from 300 to 10,000 ppm according to the manufacturer. Although the sensors come uncalibrated (at any concentration, not only at high ones), this wording is not correct. Maybe you wanted to say that **you did not calibrate** the sensor at higher concentrations, but the sensor per se is always uncalibrated from this manufacturer.

9. p9/14: "The Panterra CH4 sensor signal has been compensated for the temperature effect, but is probably not applicable for temperatures lower than 15°C." – please give the details of the sensor used in the Panterra (it is a TGS if you use the same model that I used years ago and threw away because it was unreliable); as it is, this statement is pure speculation and should either be removed or substantiated with some arguments.

**2.3 Feedback on Supplementary Information**

- in `PowerControlBoard.zip` remove the deleted file
`~$Copy of Bomexample(1).xlsx`
- in `BOM_PWCv8c_digikey.xlsx` remove unused "Sheet1"

**2.4 Technical issues**

- homogenise your variable names in text and figures (d0CH$_4$sens, d0_CH$_4$sens, d0_CH$_4$sensor; d0CH$_4$conc, d0_CH$_4$conc, d0_CH$_4$concentration,

d0 $CH_4$ concentration)
- decide whether you want to use upper case or lower case letters in figure panels
- use a space between axis title and parentheses around units
- use a degree sign where a degree sign is required (not $^0$)
- define all your variables that appear in text and figures
- spell out abbreviations upon first occurrence (e.g. AFC on p2/39)
- use some more adequate natural intervals in time axes (e.g. Fig. 5: start at midnight 00:00 and then use 3-hour intervals not some 2:24 hour intervals)
- make figure captions standalone so that the figure is understandable without reading the entire text; also define all symbols and line types (maybe a legend could help)
- do not use the term "plot" for lines or symbols in a panel of Fig. 1
- remove the erroneous and confusing superfluous ticks at the right border of panels (b) and (c) in Fig. 1
- Fig. 2: conventionally panels are labeled from left to right, then top to bottom (which would group a,b and c,d in your caption); explain what each line type indicates and what the symbols with vertical stems indicate.
- Fig. 3: add the 0 label at the beginning of the time axis; I would call this "Elapsed time" and put minute into plural
- add subscript in $CH_4$ in Fig. 3 caption and everywhere else where this was forgotten.
* * *
[Figure]

**Fig. 1.** Unpublished example of a time series with my test chamber deployed at a few ponds near Toolik Lake, Alaska, on 3 July 2015. The top panel shows the raw voltages from a TGS2600 sensor.

---

## Referee Comment (RC2) · Maciej Bartosiewicz (Referee) · 13 May 2019

This is an excellent and timely description of low-costs system allowing to monitor fluxes of CO2 and CH4 in aqautic setting with relatively high precision. I believe that this article/techincal note is suitable for publication but further care must be taken to improve the presentation. The text reads well mostly but at few places the flow should be improved. I think that even a Technical Notes will get more attention if properly streamlined. Also, and most importantly, I strongly suggest that a visual representation of the sampling system is included in the main body of the paper. I found some illustrations in the SI but a compact and clear technical scheme of the described system should be

available in the main ms. After these rather minor issues are resolved I recommend this draft for publication.

In general, and this is a strong recommendation, the graphical side of this work can/should be improved (see figures 4 and 6).

Below please find some detailed suggestions (the list is not exhaustive and probably entire text should be edited for typos and further streamlined)

Introduction

Page1 Line 35: You state that " the flux chamber method which can trap both diffusive and ebullitive (bubble) fluxes, has been demonstrated to not bias gas fluxes at the air-water interface" There is a number of publications (see Vachon et al. 2010 for example) providing evi- dence that chambers themselves can affect turbulence (and thus the flux) at the water- air interface. I also support using chambers as reference direct method for flux estima- tion but you should, at least, acknowledge that there is a potential effect on turbulence from the static chamber.

You later state that (line 1-5, page 2) "these methods are inexpensive in terms of equip-ment and work well to quantify gas emission in a confined area but they are labor intensive and have low temporal resolution" This is correct if the gas inside of the chamber is somehow homogenized. I understand that this is, in fact, tedious and adds another layer of complexity but I found it better to have a pump connected on the top of the chamber that would mix the gas inside. Otherwise, during longer deployments, $CO_2$ can accumulate inside your chamber and may bias the flux estimates.

(line 30 page 2) Using eddy covariance is well established in lakes – please reword your statement; aside from being costly and logistically complex to install, eddy co-variance datasets often require labour intensive re-processing. . . but it is certainly well established.

Methods

Specify what is the sensor in your Panthera (Neodym?), it sounds to me that these three configurations use the same methane sensor. . . "TGS2611-E00 sensor is equipped with a filter to reduce the influence of interference" what filter???? Please specify as this may be important for future use

There seems to be a problem with your bubble counter as limiting your measurements to bubbles are larger than 3-4 ml may bias total flux estimates, would it make sense to use a pre-trapping system for small bubbles?? I do not know how would/if this can be intergated but imagine a system that when CH4 bubble is detected but not quantified because of its low volume then it is directed toward another trap where cumulative volume of such small bubbles can be assessed?

You state that: "If bubbles entered the trap and were large enough to activate the venting mechanism during a non-logging period, it was missed in the logged data file" How often did this happen during your experiments and what was the size of bubbles activating venting.

---

## Author Comment (AC1) · 8 Jul 2019

Response to Reviewer 1 (Prof. Dr. Werner Eugster)

Dear Prof. Dr. Eugster,

Thank you for your detailed remarks and constructive comments which are all very helpful for improving the manuscript. We are glad that you find our work interesting and innovative. We are particularly grateful for the substantial work to understand all details, which will help us clarify the text and improve the presentation. As noted, our aim is to provide a detailed fully open-source description of the system that can contribute to more extensive data collection and to inspire technical improvements by the broader community. Our response to your specific review comments are given below.

2.1 Major issues

*2.1 Referee comment (RC)1:* The authors use three low-cost sensors, all from Figaro Inc.: the TGS2611-E00, the NGM2611-E13 (which uses the same TGS2611-E00), and the Panterra from Neodym (I assume), which in the version I used had a percursor version of the TGS2611 or so built in. I think the authors should more clearly specify (a) what sensor the Panterra uses (and provide the company names of all sensors), (b) clarify that these are (most likely) all the same sensors in different configurations (as it reads now the reader could be getting the impression that three different sensor types were tested, which is not the case)

*2.1 Author response (AR)1:* It is correct that all three sensors investigated are rather similar sensors from Figaro. The TGS2611-E00 and NGM2611-E13 differ in that the latter is attached to a small board with a potentiometer used for a crude factory calibration at 5000 ppm and with a 5-pin connector for easy plug-in to a system with a corresponding connector (cost efficient when handling many systems). The Panterra sensor is built around the Figaro TGS2610 sensor (order code PN-SM-GMT-A040A-W20A-05-R0- S0-E1-X0-I2-P0-L2-J1-Z0) from Panterra Neodym Technologies, Canada. This Panterra sensor, which was recommended after discussion with a Neodym technician, was the same as used in our study in Duc et al., 2013.

We selected sensors based on specifications and discussions with company representatives regarding several criteria, including methane specificity, a sensitivity that was potentially high enough for our applications, price and power consumption (more on this below), and believed that configuration and signal processing also can make a difference so testing different configurations was of interest to us. The sensor details were too spread out in our manuscript which we now realized was unclear.

*2.1 Author changes in manuscript (ACM)1:* The sensor details and motives behind the sensor selection will be provided in the same paragraph for all sensors together and early in the text.

*2.1 RC2:* I am concerned about the low voltages that the TGS2611-E00 gives, ca. 18–35 mV according to Fig. 3. The TGS2600 that I use delivers 200-400 mV for ambient conditions, and when I look at the specifications it appears that the manufacturer considers the TGS2611-E00 to be useful in the range of 300–10,000 ppm $CH_4$ which is way above ambient range (the TGS2600 is shown with a sensitivity to $CH_4$ in the range 1–100 ppm $CH_4$ that's the reason we selected this one for near ambient measurements in Eugster & Kling, 2012). Now there are some other publications that show that the TGS2611-E00 is actually sensitive also to near ambient conditions, but I am not yet convinced that this is the best choice for your application given the low $CH_4$ concentrations well below the range

indicated on the technical specification sheet of the manufacturer. Some more critical discussion on the sensor selection would be required in my view.

*2.1 AR2:* This question highlights and revealed an error in the script for plotting the scale of Fig. 3A. The corrected Figure 3 is pasted below. The background level of the NGM2611-E13 sensor should be in the range of few hundred millivolts, in line with your expectations.

We first intended to try the TGS2600 sensor from your study with proven high sensitivity. When contacting Figaro, they recommended we try the TGS2611-E00 and NGM2611-E13 because we wanted a high specificity for $CH_4$ and because it have higher sensitivity than specified (the Product Information notes also indicate that the sensors are far from the detection limit at the low end of the tested range; 300 ppm). We decided to follow the Figaro technician suggestions as a start and simply kept working with them because they gave an adequate response for our applications (otherwise we would have tried the TGS2600). It should be noted that previous attempts to measure absolute ambient air levels is much more demanding in terms of sensitivity than our application, which is focusing on relative change, often with a doubling or more in levels over 1-2 hours. When assessing relative changes over time in a closed system it is also important to minimize interferences of other gases that may also change over time – further explaining our sensor choice as a trade-off between sensitivity and specificity.

*2.1 ACM2:* The error in the scale of Figure 3 have been corrected and clarifications on sensor details and motives behind the sensor selection will be made (see ACM1 above).

[Figure]

*2.1 RC3:* The authors only sample data every minute, which I find utterly coarse. They may have a reason for this, but in my own tests with the TGS2600 a one-minute measurement interval in combination with a 5-minute data rejection after chamber deployment (page 6, line 26; this information should actually have been given in the Methods section already, because this is an essential flaw in the system in my view) I would have lost all the information relevant to chamber

fluxes (see graph below and description of unpublished internal example graph from my experiments at Toolik Lake, Alaska, USA). Thus, the authors should more precisely describe their method and critically discuss such shortcomings to help others to do better.

*2.1 AR3:* One goal of our project was to develop an active wireless sensor network in which data from many flux chambers (called clients) are sent by Xbee radio transmitter modules and recorded on a small low-cost Raspberry Pi computer on the lakeshore. The sampling data rate, so far, is constrained to maintain the digimesh network working with minimum labor effort. In bad weather conditions, radio communication is easily broken, therefore it can take some minutes to re-establish the communication depending on the distance of the clients. During this offline period, the limited memory buffer of these data loggers does not allow us to sample very often. A second reason for the one-minute measurement internal is driven by coordinated logging and transmitting data from a $CO_2$/RH/Temp sensor having a 25 second measurement cycle – which also presently restricts the measurement frequency.

The rejection of data from the initial 5-minutes of measurements, is because the $CH_4$ sensor signal can be affected by temperature and relative humidity, and when focusing on relative change it is again important to minimize the influence of other confounding factors that may change over time. This rejection period is mainly to wait for temperature and relative humidity in the chamber to stabilize after chamber closure on the water surface to ensure that the $CH_4$ sensor response reflects $CH_4$ and not changing humidity and or temperature.

From your attached graph, it appears that the temperature and relative humidity in your chamber reach stable equilibrium quite quickly. In our case, we use the temperature and relative humidity measured from an integrated sensor onboard the $CO_2$ sensor which is measured every minute along with $CO_2$ concentration. Of course, an obvious data interpretation improvement would be to modify the length of the initial period during which data is not used to the actual time it takes to reach stable enough relative humidity and temperature, instead of having the static 5-minute period used here for simplicity.

In spite of our low measurement frequency, our results show that the system is still able to capture relative changes adequately. Of course, the situation is very different in applications aiming for accurate absolute levels in ambient air, and for such applications a high measurement frequency is of course more important to cancel out sensor noise in data processing than in our flux chamber application.

*2.1 ACM3:* The explanations and motivations provided above in AR3 will be clarified in the manuscript in a discussion paragraph devoted to measurement frequency.

*RC4:* The ebullition (bubble) counter is quite interesting, but with a bubble volume of 3-4 mL required to actually leave a defensible signal, this does not yet seem to be an optimum choice. Here, a reference to and comparison with the (commercial) system of Andreas Mäck (doi:10.5194/bg-11-2925-2014) would be helpful. Since the Varadharajan et al. (2010) reference (pages 2,3,8, lines 30,22,21) is not listed in the References, I could not convince myself that this bubble counter system is really thoroughly tested and reliable. In the discussion you only say "For a long term solution, the recent study using optical sensors in an open path funnel (Delwiche and Hemond, 2017) 30 suggests an alternative and interesting design for ebullition studies, which could be combined with the present sensor approach to also quantify $CH_4$ content in the bubbles." – thus does this mean that you are satisfied with the performance for short-term investigations? I am not really convinced and would appreciate a somewhat clearer statement what your recommendation is for studies that are shorter than a "long-term solution".

*2.1 AR4:* We apologize for the missing Varadharajan et al. (2010) reference and will correct this. We were not aware of the Meack et al 2014 paper and will integrate this to the manuscript.

Most previous bubble counter systems are based on bubble volume quantification by differential pressure sensors (e.g. Varadharajan et al. 2010; Maeck et al 2014) or optical sensors (Delwiche and Hemond, 2017). The detection limit of the differential pressure measurement, in our case corresponding to 3-4 ml gas, depends on the shape of the cylinder where the bubbles accumulate. Therefore, the longer and narrower a cylinder, the lower the detection limit. In turn, this leads to a trade-off, where the more sensitive systems become too tall for deployment in shallow waters, which often have proportionally higher ebullition rates. Our detection limit was chosen to allow deployment in shallow water. The shorter funnel of Delwiche and Hemond, 2017 (based on the optical sensor) cobined with our system could solve the challenge we face deploying in shallow waters. We apologize that this statement was not clear in the manuscript.

*2.1 ACM4:* The response above including explanation of trade-offs and choices will be clarified in the revised manuscript. We will also more clearly relate to similar studies and have added proper references to Varadharajan et al 2010 and Maeck et al 2014.

*2.1 RC5:* Your regressions (I assume you use ordinary least-squares regressions) are not correct from a statistical viewpoint (Figs 4 and 5): you must reverse the dependent and the independent variable: you want to find out how to use the signal to compute the true concentration using your regressions, not the other way around (i.e. to predict the signal based on knowledge of the concentration – that's what your regressions show).

*2.1 AR5:* Figure 4 is to be seen as a calibration curve where the $CH_4$ concentrations are measured by reference instruments (GC-FID or an LGR greenhouse gas analyzer) and thereby represents the independent (x-axis) data. The $CH_4$ sensor response in the calibration case becomes the dependent signal. We agree that in a case when wanting to calculate concentrations from sensor signals, reversing the dependent and independent variables, would make more sense.

Figure 5 is just showing the sensor signal over time and simply provide temporal information of multiple variables (time on x-axis) and no regressions are made directly in this graph.

*2.1 AMC5:* A clarification that Figure 4 represents a calibration curve will be added.

*2.1 RC6:* I always use a fan in chambers, you don't. I understand that this corresponds to some static chambers that people use with syringe sampling, but in your case I am concerned that without a fan to mix the volume of the chamber the $CO_2$ (which is heavier than air) starts to accumulate above the water surface, and then a steep gradient creeps upwards where I expect your sensors; hence this linear increase in Fig. 5. Contrastingly, $CH_4$ which is lighter than air, quickly would accumulate under the top of the chamber, and hence probably the curvature although I would have expected that the $CO_2$ saturation should occur earlier than the $CH_4$ saturation in such a chamber. Please comment on this and justify why not to mix the air inside the chamber to ascertain representative concentration measurements inside the flux chamber.

*2.1 AR6:* Our floating chamber is light-weight and freely moves up and down with the water. There is also some wind induced drifting around the separate anchored float. The sensors in the chamber are about 10 cm above water surface. We believe that the natural movement of chamber caused by winds and waves mixes the volume in the chamber.

We are aware of the vital need of air mixing in chambers holding vegetation, but for open water cases we see a risk that adding a fan will create an unknown bias from the added fan-induced turbulence and weight.

Some chambers designed differently have reported biased fluxes, while the chamber design used here without fans have repeatedly shown negligible bias compared to non-invasive techniques under variable conditions ranging from coastal water, small lakes and streams (Cole at al 2010; Gålfalk et al 2013; Lorke et al. 2015). Hence, we preferred to keep this tested chamber design.

*2.1 ACM6:* The choice of the chamber design and evidence in support of it will be clarified.

*2.1 RC7:* According to the manufacturers information the sensor resistance decreases as the $CH_4$ concentration increases, thus in principle the voltage you measure should decrease not increase with increasing $CH_4$ – but your measurements in Fig. 4 show the exact opposite of what one would expect from the manufacturer information. Do you have an explanation for this? I must admit that on short timescales I see the same (see Figure inserted below), but on longer timescales I see what I would expect from the manufacturer's data sheet. Our procedure suggested by Eugster & Kling (2012) solve this issue with the calibration – after linearisation the use of a high and low calibration point simply reverses the sign if the response is of the kind that you show in Fig. 4. If you have an explanation why the TGS2611-E00 has increasing voltage with increasing $CH_4$ concentration then this would be a helpful insight for the reader. If you don't have an explanation, maybe you have an opinion?

*2.1 AR7:* From our understanding, the NGM2611-E00 and the TGS 2611 sensor in a circuit essentially operates as a voltage divider. A figure and associated text from the Product Information note of TGS 2611 (https://www.figaro.co.jp/en/product/docs/tgs2611_product_information_rev02.pdf) is provided below. The resistance of the $CH_4$ sensor is called Rs which has a resistance value that decreases as $CH_4$ concentration increases, load resistor $R_L$ has constant value (about 5 kΩ). This circuit is fed by a constant circuit voltage $V_C$ (5V), and the current equals to ratio of $5/(Rs+R_L)$. Hence, if Rs decreases, the current will increase. As a result, output voltage ($V_{RL}$) which equals to $R_L*5/(Rs+R_L)$ will increase. We have not studied the circuit details for other sensors so perhaps interpretations of the output voltage differ among sensors which may be the reason for this comment. We will double check our understanding and if it seems OK we will try to clarify this in the supplementary material.

[Figure]

**Basic Measuring Circuit:**

The sensor requires two voltage inputs: heater voltage ($V_H$) and circuit voltage ($V_C$). The heater voltage ($V_H$) is applied to the integrated heater in order to maintain the sensing element at a specific temperature which is optimal for sensing. Circuit voltage ($V_C$) is applied to allow measurement of voltage ($V_{RL}$) across a load resistor ($R_L$) which is connected in series with the sensor.

A common power supply circuit can be used for both $V_C$ and $V_H$ to fulfill the sensor's electrical requirements. The value of the load resistor ($R_L$) should be chosen to optimize the alarm threshold value, keeping power dissipation ($P_S$) of the semiconductor below a limit of 15mW. Power dissipation ($P_S$) will be highest when the value of Rs is equal to $R_L$ on exposure to gas.

*2.1 ACM7:* The above explanation will be clarified in the supplementary material.

2.2 Minor Issues

*2.2 RC1:* p2/33: "The eddy covariance (EC) technique is increasingly used for long-term monitoring, but it is expensive in terms of equipment and is still being evaluated for aquatic environments." – what do you mean with this statement? I don't consider this to be correct, the method is in use beyond evaluation. Please provide some references and reword the second part. For example, we have authored a couple papers and also written a chapter in the book Eddy Covariance: A Practical Guide to Measurement and Data Analysis (chapter on lakes: doi:10.1007/978-94-007-2351-1_15). Thus, the method is established (at least better than your chambers, to be more direct) – but I agree that it is costly and I agree that such low-cost sensors are important.

*2.2 AR1:* We did not intend to unfairly describe EC measurements. Given the discussions on issues such as lateral fluxes (land/sea breeze effects), wind shadow zones around forested lake shores, other irregularities in wind patterns over lakes, challenges interpreting footprint locations and shape for small lakes, and other discussions on suitable equipment (e.g. open or closed path gas analyzers), we simply had the impression that method evaluation and development was still ongoing. Because the referee comment clearly signals we were wrong we will remove this statement.

*2.2 ACM1:* We will reword the sentence to: "The eddy covariance (EC) technique is increasingly used for long-term monitoring of terrestrial and lake-dominated landscapes, but it is expensive in terms of equipment."

*2.2 RC2: p2/45: " The CH4 sensor tested here is a Taguchi Gas Sensor (TGS) (Figaro Engineering Inc.,* Osaka, Japan). It is a high sensitivity $CH_4$ gas sensor. . . ": I completely disagree, it is a low-sensitivity sensor which (in the version you use) only has a manufacturer specified lower measurement range of 300 ppm $CH_4$! I already realize that our more cautious wordings about the TGS2600 (which has a higher sensitivity than the TGS2611-E00) is ignored by some others, which can lead to frustration. Be clear that this is experimental work trying to squeeze the tiny bit of information out of a sensor that is not made for ambient concentrations – but I agree that it has some value for such measurements.

*2.2 AR1:* We meant that the sensor is more sensitive than several other $CH_4$ sensors in the same prize class, but we agree with the reviewer and now see that the statement can be interpreted in misleading ways.

*2.2 ACM2:* We will remove this statement (See also AR2.1, AR2)

*2.2 RC3.* You never specified which pressure sensor you used, thus it is unclear to me why you did not use an I2C sensor, there plenty of those on the market. What is the special advantage of your pressure sensor that requires an AD620 amplifier to be useful? This remains obscure to the reader.

*2.2 AR3* This manuscript describes work that is a follow up from a previous study on an automatic system to measure greenhouse gases from aquatic environments (Duc et al., 2013). In our previous electronic circuit, we used an AD620 amplifier for the pressure sensor (26PCDFA6G, Honeywell, Sensing and Control, Canada) to measure atmospheric pressure. After reading the work of Varadharajan et al. (2010), we adapted our system by simply changing one external resistor to get the proper gain factor to use with our pressure sensor (26PCAFA6D).

*2.2 ACM3:* We will add the sensor information and clarify our reasons for choosing this sensor to the manuscript.

*2.2 RC4:* Eq. 1 should use SI units or at least the same units of the same physical quantity and not include obscure conversion factors. Thus, you must decide whether your time variables should be in hours or in seconds (the primary SI unit) or minutes, please no mixtures.

*2.2 AR4:* The flux time unit was in hours representing a relevant time unit given the accumulation time of the chamber. As this study focuses on evaluating the sensor response to the change in mixing ratio of $CH_4$ and $CO_2$ gases in the chamber in which the sensor signal is recorded every minute, we have decided to present $\Delta C/\Delta t$ (ppmv min$^{-1}$) to avoid applying a conversion factor.

*2.2 ACM4:* The above explanation will be provided in the manuscript.

*2.2 RC5, p5/25*: check your instrument information, most likely this was an LGR FGGA (not a DLT-100, which as I remember is a $CH_4$-only instrument) that measures $CH_4$ and $CO_2$.

*2.2 AR5:* The reviewer is correct - instrument we used is an early benchtop version of the FGGA analyzer that have the capacity to measure $CH_4$, $CO_2$ and $H_2O$. It has DLT-100 printed on the cover leading to this confusion.

*2.2 ACM5:* We will edit the manuscript to include "a FGGA with capacity to measure $CH_4$, $CO_2$ and $H_2O$."

*2.2 RC6, p6/8:* you did not specify what your "baseline noise" actually is. Is it the square root of the variance or the noise baseline derived from an Allan variance plot, or anything else? Some more details in the Methods section would be really helpful.

*2.2 AR6:* Our baseline noise is square root of the variance.

*2.2 ACM6:* We will add this clarification to the manuscript.

*2.2 RC7 p6/13:* "The pressure in the trap was affected by air temperature, especially the diel temperature cycle." – this sounds like an error, pressure is a physical entity that is independent of temperature, thus this must be wrong. What I can imagine is that you mean that your pressure sensor signal (but not the pressure itself) depended on air temperature. Please correct.

*2.2 AR7:* We did mean to refer to the pressure sensor signal.

*2.2 ACM7:* We will correct this in our manuscript to read "The pressure sensor signal measured in the trap was affected…."

*2.2 RC8 p8/39:* "However, since the sensor is not calibrated for very high concentrations, we could not determine the flux rate observed during these events." – I completely disagree, at least for $CH_4$ (you do not really reveal any necessary details on the $CO_2$ measurements . . . ): the TGS2611-E00 has a specified measurement range from 300 to 10,000 ppm according to the manufacturer. Although the sensors come uncalibrated (at any concentration, not only at high ones), this wording is not correct. Maybe you wanted to say that you did not calibrate the sensor at higher concentrations, but the sensor per se is always uncalibrated from this manufacturer.

*2.2AR8:* Correct and thanks. We did not calibrate the sensor at higher concentrations.

*2.2 ACM8:* We will edit this statement in the manuscript to read "However, since we did not calibrate the sensor for high concentrations, we could not determine the flux rate….".

*2.2 RC9 p9/14:* "The Panterra $CH_4$ sensor signal has been compensated for the temperature effect, but is probably not applicable for temperatures lower than 15°C." – please give the details of the sensor used in the Panterra (it is a TGS if you use the same model that I used years ago and threw away because it was unreliable); as it is, this statement is pure speculation and should either be removed or substantiated with some arguments.

*2.2AR9:* The Panterra sensor product features identifies that it has active temperature compensation (http://neodymsystems.com/download/Panterra-MOS-ALL_Brief_101.pdf). We did try calibrating at temperatures lower than 15°C but still the sensor deviated in its response from other TGS sensors. We did not do further testing to investigate this response because it seemed more efficient to focus on the other sensors.

*2.2 ACM9:* We will remove this sentence from the manuscript as we did not investigate this further.

2.3 Feedback on Supplementary Information
*2.3 RC1:* in PowerControlBoard.zip remove the deleted file
~$Copy of Bomexample(1).xlsx

*2.3 ACM1:* We will remove this unused file.

*2.3 RC2:* in BOM_PWCv8c_digikey.xlsx remove unused "Sheet1"

*2.3 ACM2:* We will remove "Sheet1".

2.4 Technical issues
- homogenise your variable names in text and figures (d0CH$_4$sens, d0_CH$_4$sens, d0_CH$_4$sensor; d0CH$_4$conc, d0_CH$_4$conc, d0_CH$_4$concentration, d0CH$_4$concentration)
- decide whether you want to use upper case or lower case letters in figure panels
- use a space between axis title and parentheses around units
- use a degree sign where a degree sign is required (not 0)
- define all your variables that appear in text and figures
- spell out abbreviations upon first occurrence (e.g. AFC on p2/39)
- use some more adequate natural intervals in time axes (e.g. Fig. 5: start at midnight
00:00 and then use 3-hour intervals not some 2:24 hour intervals)

- make figure captions standalone so that the figure is understandable without reading the entire text; also define all symbols and line types (maybe a legend could help)
- do not use the term "plot" for lines or symbols in a panel of Fig. 1
- remove the erroneous and confusing superfluous ticks at the right border of panels (b) and (c) in Fig. 1
- Fig. 2: conventionally panels are labeled from left to right, then top to bottom (which would group a,b and c,d in your caption); explain what each line type indicates and what the symbols with vertical stems indicate.
- Fig. 3: add the 0 label at the beginning of the time axis; I would call this "Elapsed time" and put minute into plural
- add subscript in $CH_4$ in Fig. 3 caption and everywhere else where this was forgotten.

*2.4 ACM1:* We will update the manuscript to include all the above changes.

References:

Duc, N. T., Silverstein, S., Lundmark, L., Reyier, H., Crill, P., and Bastviken, D.: Automated Flux Chamber for Investigating Gas Flux at Water–Air Interfaces, Environ. Sci. Technol., 47, 968-975, 10.1021/es303848x, 2013.

Cole, J.J., Bade, D.L., Bastviken, D., Pace, M.L. and Van de Bogert, M. (2010) Multiple approaches to estimating air-water gas exchange in small lakes. Limnology and Oceanography-Methods 8, 285-293.

Gålfalk, M., Bastviken, D., Fredriksson, S. and Arneborg, L. (2013) Determination of the piston velocity for water-air interfaces using flux chambers, acoustic Doppler velocimetry, and IR imaging of the water surface. Journal of Geophysical Research: Biogeosciences 118, 770-782.

Lorke, A., Bodmer, P., Noss, C., Alshboul, Z., Koschorreck, M., Somlai-Haase, C., Bastviken, D., Flury, S., McGinnis, D.F., Maeck, A., Muller, D. and Premke, K. (2015) Technical note: drifting versus anchored flux chambers for measuring greenhouse gas emissions from running waters. Biogeosciences 12, 7013-7024.

Maeck, A., Hofmann, H., and Lorke, A.: Pumping methane out of aquatic sediments – ebullition forcing mechanisms in an impounded river, Biogeosciences, 11, 2925-2938, https://doi.org/10.5194/bg-11-2925-2014, 2014.

Varadharajan, C., Hermosillo, R., and Hemond, H. F.: A low-cost automated trap to measure bubbling gas fluxes, Limnology and Oceanography: Methods, 8, 363-375, 2010.

---

## Author Comment (AC2) · 8 Jul 2019

Response to Reviewer 2 (Dr. Maciej Bartosiewicz)
Dear Dr. Bartosiewicz,

Many thanks for your valuable comments and detailed remarks on our manuscript and your suggestions to improve the system. We very much appreciate your review efforts that will be very helpful in improving our manuscript.

As noted earlier, we aim for providing a detailed, fully open-source description of the system that can contribute to more extensive data collection and to inspire technical improvements by the broader community. Our response to your specific review comments are given below.

*1.0 RC 1*: This is an excellent and timely description of low-costs system allowing to monitor fluxes of CO2 and CH4 in aqautic setting with relatively high precision. I believe that this article/techincal note is suitable for publication but further care must be taken to improve the presentation. The text reads well mostly but a few places the flow should be improved. I think that even a Technical Notes will get more attention if properly streamlined. Also, and most importantly, I strongly suggest that a visual representation of the sampling system is included in the main body of the paper. I found some illustrations in the SI but a compact and clear technical scheme of the described system should be available in the main ms. After these rather minor issues are resolved I recommend this draft for publication.

*1.0 AR1*: Given that this type of manuscript is expected to be short we decided to move the visual representation to the supplemental information (SI) section.

*1.0 ACM1*: We will discuss with the editor if we can add a more substantial visual representation of the sampling system to the main body of the paper.

*1.0 RC2 Page1 Line 35*: You state that " the flux chamber method which can trap both diffusive and ebullitive (bubble) fluxes, has been demonstrated to not bias gas fluxes at the air water interface" There is a number of publications (see Vachon et al. 2010 for example) providing evidence that chambers themselves can affect turbulence (and thus the flux) at the water- air interface. I also support using chambers as reference direct method for flux estimation but you should, at least, acknowledge that there is a potential effect on turbulence from the static chamber.

*1.0 AR2:* Clear turbulence effects have been noted for some chambers designs/approaches, such as chambers attached to a stationary and heavy objects (e.g. ADV instrument in the mentioned reference), while the chamber design used here, with light weight, limited intrusion of chamber walls into the water, and chamber mooring to enable the chamber to following wave or water motion as much as possible, has repeatedly shown negligible bias compared to non-invasive techniques under variable conditions ranging from coastal water, small lakes and streams (Cole at al 2010; Gålfalk et al 2013; Lorke et al. 2015).

*1.0 ACM2:* As we have stated in Response to Reviewer 1 comments *2.1 ACM6:* The choice of the chamber design and evidence in support of it will be clarified. In this clarification we will show awareness of the possible turbulence effects as suggested here.

*1.0 RC3 (line 1-5, page 2):* You later state that (line 1-5, page 2) "these methods are inexpensive in terms of equipment and work well to quantify gas emission in a confined area but they are labor intensive and have low temporal resolution" This is correct if the gas inside of the chamber is somehow homogenized. I understand that this is, in fact, tedious and adds another layer of complexity but I found it better to have a pump connected on the top of the chamber that would mix the gas

inside. Otherwise, during longer deployments, $CO_2$ can accumulate inside your chamber and may bias the flux estimates.

*1.0 AR3:* In our statement "*these methods are inexpensive in terms of equipment and work well to quantify gas emission in a confined area but they are labor intensive and have low temporal resolution* ", we were referring to the entire sampling process which in most sampling setups on a lake requires traveling (often rowing) out to deploy and then returning some time later to sample and returning samples to the laboratory for analyses. We agree also that adding the manual mixing of the headspace also adds time.

*1.0 ACM3:* We will correct the sentence in the manuscript to read "*Both these methods are inexpensive in terms of equipment and work well to quantify gas emission in a confined area however they are labor intensive due to repeated visits for both deployment and sample collection and therefore often have low temporal resolution.*"

*1.0 RC4 (line 30 page 2):* Using eddy covariance is well established in lakes – please reword your statement; aside from being costly and logistically complex to install, eddy covariance datasets often require labour intensive re-processing. . . but it is certainly well established.

*1.0 AR4:* As in our response to reviewer 1 comments 2.2 AR 1, we did not intend to unfairly describe EC measurements. Given the discussions on issues such as lateral fluxes (land/sea breeze effects), wind shadow zones around forested lake shores, other irregularities in wind patterns over lakes, challenges interpreting footprint locations and shape for small lakes, and other discussions on suitable equipment (e.g. open or closed path), we simply had the impression that method evaluation and development was still ongoing. Because the referee comment clearly signals we were wrong we will remove this statement.

*1.0 ACM4:* We will reword the sentence to: "The eddy covariance (EC) technique is increasingly used for long-term monitoring of terrestrial and lake-dominated landscapes, but it is expensive in terms of equipment."

*1.0 RC5:* Specify what is the sensor in your Panthera (Neodym?), it sounds to me that these three configurations use the same methane sensor. . . "TGS2611-E00 sensor is equipped with a filter to reduce the influence of interference" what filter???? Please specify as this may be important for future use.

*1.0 AR5:* The manufacturer does not provide information about the filter in the product description only that it is meant to "eliminate the influence of interference gases such as alcohol, resulting in highly selective response to methane gas".

*1.0 ACM5:* We will modify the sentence to include this information from the manufacturer: "The TGS2611-E00 sensor is equipped with a filter to eliminate the influence of interference gases such as alcohol, resulting in a selective response to $CH_4$.". and we will add the manufacturer's product website (http://www.figarosensor.com/product/entry/tgs2611-e00.html) as a reference.

*1.0 RC6:* There seems to be a problem with your bubble counter as limiting your measurements to bubbles are larger than 3-4 ml may bias total flux estimates, would it make sense to use a pre-trapping system for small bubbles?? I do not know how would/if this can be integrated but imagine a system that when $CH_4$ bubble is detected but not quantified because of its low volume then it is directed toward another trap where cumulative volume of such small bubbles can be assessed?

*1.0 AR6:* We appreciate the suggestion for modification to pre-trap small bubbles. We decided on this design of an "accumulation trap" so that all the bubbles will be trapped, but the limitation in technology and the variable conditions of deployment outdoors does not allow us to resolve very small bubbles.

*1.0 ACM6:* We will clarify that ours system accumulates all bubbles and perform the measurements when the bubble volume is large enough. Hence the limited resolution of small bubbles does not mean that the small bubbles are missed but more that the pressure sensor measurements do not resolve individual small bubbles but instead measure the average volume of many bubbles.

*1.0 RC7:* You state that: "If bubbles entered the trap and were large enough to activate the venting mechanism during a non-logging period, it was missed in the logged data file" How often did this happen during your experiments and what was the size of bubbles activating venting.

*1.0 AR7:* The upper threshold of the bubble trap is a technical feature of the trap design however we have not observed this during any of our tests. In this study the accumulated bubble volume of 28 ml will activate the venting, and we did not experience any bubble events larger than this threshold, but we think it is important to report this setting and the need to consider it to potential users.

*1.0 ACM7:* We will add a statement to clarify this in the manuscript.

References:

Cole, J.J., Bade, D.L., Bastviken, D., Pace, M.L. and Van de Bogert, M. (2010) Multiple approaches to estimating air-water gas exchange in small lakes. Limnology and Oceanography-Methods 8, 285-293.

Gålfalk, M., Bastviken, D., Fredriksson, S. and Arneborg, L. (2013) Determination of the piston velocity for water-air interfaces using flux chambers, acoustic Doppler velocimetry, and IR imaging of the water surface. Journal of Geophysical Research: Biogeosciences 118, 770-782.

Lorke, A., Bodmer, P., Noss, C., Alshboul, Z., Koschorreck, M., Somlai-Haase, C., Bastviken, D., Flury, S., McGinnis, D.F., Maeck, A., Muller, D. and Premke, K. (2015) Technical note: drifting versus anchored flux chambers for measuring greenhouse gas emissions from running waters. Biogeosciences 12, 7013-7024.